# ⊕CHROKNOWLEDGE: UNVEILING CHRONOLOGICAL KNOWLEDGE OF LANGUAGE MODELS IN MULTIPLE DOMAINS

**Yein Park**[1], **Chanwoong Yoon**[1], **Jungwoo Park**[1,3],
**Donghyeon Lee**[1,3], **Minbyul Jeong**[2*], **Jaewoo Kang**[1,3*]
Korea University[1]    Upstage AI[2]    AIGEN Sciences[3]
{522yein, cwyoon99, jungwoo-park, dong9733, kangj}@korea.ac.kr

## ABSTRACT

Large language models (LLMs) have brought significant changes to many aspects of our lives. However, assessing and ensuring their chronological knowledge remains challenging. Existing approaches fall short in addressing the temporal adaptability of knowledge, often relying on a fixed time-point view. To overcome this, we introduce **CHROKNOWBENCH**, a benchmark dataset designed to evaluate chronologically accumulated knowledge across three key aspects: multiple domains, time dependency, temporal state. Our benchmark distinguishes between knowledge that evolves (e.g., personal history, scientific discoveries, amended laws) and knowledge that remain constant (e.g., mathematical truths, commonsense facts). Building on this benchmark, we present **CHROKNOWLEDGE** (Chronological Categorization of Knowledge), a novel sampling-based framework for evaluating LLMs' non-parametric chronological knowledge. Our evaluation led to the following observations: (1) The ability of eliciting temporal knowledge varies depending on the data format that model was trained on. (2) LLMs partially recall knowledge or show a cut-off at temporal boundaries rather than recalling all aspects of knowledge correctly. Thus, we apply our **CHROKNOWPROMPT**, an in-depth prompting to elicit chronological knowledge by traversing step-by-step through the surrounding time spans. We observe that it successfully recalls objects across both open-source and proprietary LLMs, demonstrating versatility, though it faces challenges with dynamic datasets and unstructured formats.[1]

## 1 INTRODUCTION

Do large language models (LLMs) possess the ability to understand and track the history of knowledge as time progresses? In other words, can these models, which represent the cutting edge of modern artificial intelligence, reason appropriately about questions that involve evolving facts? Although some details remain controversial, knowledge—like science—is built upon accumulation (Zeigler, 2012; Picho et al., 2016). From raw data to information and to knowledge, every bit is cumulative which contributes to progress across all domains. This accumulation forms the foundation for higher-level reasoning, which is akin to wisdom in navigating the complexities of our world (Rowley, 2007). Given that LLMs are trained on vast and diverse corpora and are now integral to numerous applications in our daily lives, they must remain accurate and up-to-date to ensure reliability. Early versions of ChatGPT (OpenAI, 2022), for instance, sometimes produced inaccurate or absurd responses like the infamous example of "*The happening of King Sejong (1397-1450) throwing MacBook (2016-)*"[2]. These errors still give us a lesson that we need more precise model recalling knowledge correctly.

When we examine the issue closely, it's not just a matter of hallucination but also about whether the alignment of knowledge, particularly regarding dates, is accurate. Ensuring that LLMs maintain current and contextually relevant knowledge over time is crucial and researchers have explored

---

*Corresponding authors
[1]Our datasets and code are publicly available at https://github.com/dmis-lab/ChroKnowledge
[2]https://english.hani.co.kr/arti/english_edition/e_international/1095956

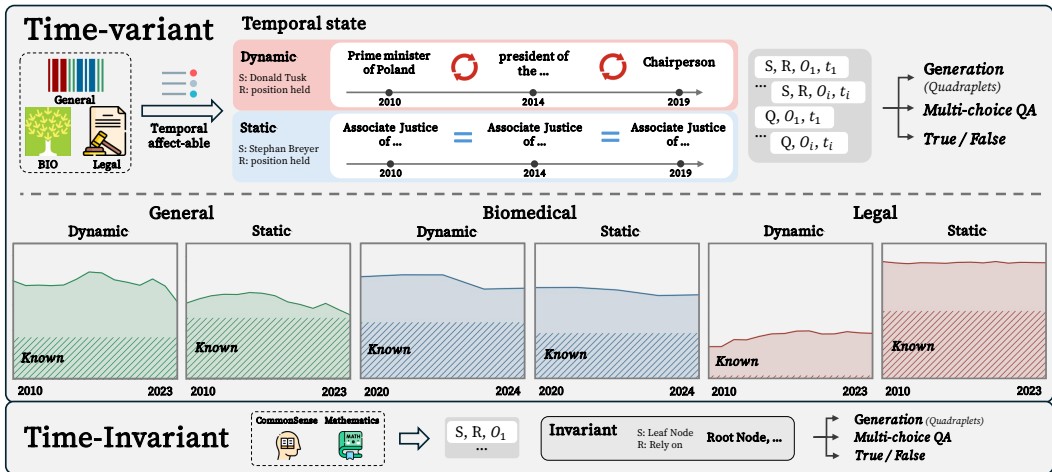

Figure 1: The overview of **ChroKnowBench**. We gather knowledge with time stamps and separate them in three key aspects: (1) multiple domains: general, biomedical, legal, commonsense, and mathematics; (2) time dependency: as time goes by, changeable knowledge or not; (3) temporal state: dynamic (has evolved over period) and static (no change occurred during period). Here, trends of *Correct* (§2.1) for each years represented by line plots show difference among domains and temporal states. And each highlighted portions are chronologically ***Known*** following §6.1.

various ways to investigate and verify the knowledge within these models (Zhang et al., 2024b). Pioneering works investigate whether language model has an ability of knowledge base or not in diverse domains (Petroni et al., 2019; Sung et al., 2021). Many subsequent studies analyze how LLMs define knowledge (Dai et al., 2022; Mishra et al., 2024); exploit how LLMs represent their knowledge (Geva et al., 2023; Zheng et al., 2024) with temporal context (Kasai et al., 2023; Fatemi et al., 2024); edit the misleading aspects of knowledge (Manakul et al., 2023; Wang et al., 2024c).

Here, we raise a question: "Do these methods sufficiently address the temporal adaptability of knowledge?". Current temporal-related approaches for evaluating and updating LLMs often focus on single time stamps, struggling to address the adaptable characteristics of knowledge over time (Jang et al., 2022a; Ge et al., 2024), which are especially important in specialized domains such as scientific discoveries and amended laws. This limitation can lead to outdated or incomplete information, undermining the models' effectiveness and safety. Addressing these challenges requires a comprehensive approach—temporal evolution, a core component of the knowledge accumulation.

Thus, we introduce **CHROKNOWBENCH**, a benchmark dataset designed to evaluate chronologically accumulated knowledge across three dimensions: time dependency (time variant and time invariant), multiple domains (general, biomedical etc.), and temporal state (dynamic and static). CHROKNOWBENCH differentiates between knowledge that is subject to evolution (e.g., transfer situation of a soccer player, scientific discoveries, and amended legal regulations)—focusing on transformations in object-specific attributes such as roles or affiliations while keeping the subject and relation fixed—and knowledge that remains invariant (e.g., mathematical truths and commonsense facts). This object-level focus allows for precise and interpretable assessments of temporal knowledge dynamics by isolating changes in object-specific attributes. We then classify domains based on whether they are influenced by the flow of time, considering the domain specificity. Finally, we set the time frame to categorize the knowledge as either changeable or steady (Section 3).

Building on this benchmark, we also present **CHROKNOWLEDGE** (Chronological Categorization of Knowledge), a novel framework for assessing and enhancing the non-parametric chronological knowledge of LLMs. So, we start from analyzing how current open-source and proprietary LLMs work. As we expected, the time invariant knowledge shows steady in all time frame. However, for time variant dataset, the domain-specific characteristics significantly influence the representation of temporal knowledge from LLMs. More stable domains exhibit consistent performance, while more variable domains show more fluctuations. These observations highlight that we need a comprehensive approach to enhance representing temporal knowledge from LLMs (Section 5).

To this end, our CHROKNOWPROMPT approach utilizes an in-depth chronological prompting strategy that traverses knowledge across adjacent time spans, effectively addressing issues of partial recall and

Table 1: Knowledge categorization with a temporal component. We classify responses into *Correct*, *Partial Correct*, and *Incorrect* to specify eliciting predictions in diverse way by comparing them with the answer set $A$. We use a temperature set $\mathcal{T} \in {0, 0.7}$ to capture variations in prediction, where $\mathcal{T}$ includes both greedy decoding and temperature sampling. We set $n$ as 5, meaning that we evaluate using five distinct combinations of few-shot exemplars to ensure the robust assessment.

| Category | Definition | Description |
|---|---|---|
| Correct | $\{\hat{o}_i \mid M(D_i, s, r, t) = \hat{o}_i; M, \tau = 0\}_{i=1}^n \subseteq A$ | All objects generated with greedy decoding are entirely included within the answer set. |
| Partial Correct | $\bigcup_{\tau \in \mathcal{T}} \{\hat{o}_i \mid M(D_i, s, r, t) = \hat{o}_i; M, \tau\}_{i=1}^n \cap A \neq \emptyset$ | At least one generated object from greedy decoding or temperature sampling is in the answer set. |
| Incorrect | $\bigcup_{\tau \in \mathcal{T}} \{\hat{o}_i \mid M(D_i, s, r, t) = \hat{o}_i; M, \tau\}_{i=1}^n \cap A = \emptyset$ | None of the generated objects, either from greedy decoding or temperature sampling, are included in the answer set. |

temporal boundaries (Section 6). In knowledge recall, our evaluation reveals improvements in the biomedical domain (11.5%) and the general domain (2.8%), shifting knowledge category from *Partial Known* to *Known* for unchanged objects. Our non-parametric approach allows for direct updates to both proprietary and open-source LLMs without extensive retraining, highlighting practicality and broad applicability (Section 7). Our work emphasizes the importance of temporal context in eliciting LLMs knowledge while identifying challenges with dynamic datasets and unstructured, context-rich formats like in the legal domain. A comprehensive analysis advocates for integrating parametric techniques to complement prompting and achieve more robust temporal knowledge handling.

## 2 PRELIMINARIES

### 2.1 KNOWLEDGE CATEGORIZATION WITH A TEMPORAL COMPONENT

To distinguish and evaluate the knowledge levels of language models, we utilize the Sampling-based Knowledge Categorization (SliCK) framework (Gekhman et al., 2024). This approach starts by sampling the model $M$'s answers to questions using various few-shot exemplar sets $D$. The sampling is conducted under two temperature conditions: $\tau = 0$ and $\tau > 0$. Then, it categorizes the degree to which the model knows each piece of knowledge into four levels: *HighlyKnown, MaybeKnown, WeaklyKnown*, and *Unknown*.

Based on Gekhman et al. (2024), we make the following modifications as follows: (1) We append a temporal component $t$ to the conventional $\{subject\ (s), relation\ (r), object\ (o)\}$ triplet structure, allowing us to evaluate the model's knowledge across different time stamps; (2) We merge the two categories (*MaybeKnown* and *WeaklyKnown*) that represent recallable knowledge states[3] into a single category (*Partial Correct*); (3) By using time attribute, we also renamed the *HighlyKnown* and *Unknown* to the *Correct* and *Incorrect*, respectively. Our detailed definitions and descriptions are provided in Table 1. Although this setting allows us to categorize the model's sampled responses more precisely regarding time attribute, it only captures the model's knowledge at specific time points $t$, limiting our ability to observe changes over time. We address this limitation in Section 6.

### 2.2 ELICITING KNOWLEDGE USING DIVERSE TEMPLATES

Since models prefer different formats when eliciting their knowledge, it is important to use varied approaches to accurately assess their understanding (Zhou et al., 2024). While we initially evaluate the model's knowledge using a standard triplet format, relying on a single template may not sufficiently capture the full extent of the model's knowledge. Thus, following Hendrycks et al. (2021); Huang et al. (2024), we also employ a well-known format, multiple-choice question answering (MCQA) with 4 options, and True/False to elicit the model's knowledge more effectively. As a result, we propose three templates for measuring how much knowledge the model holds: triplets (hereafter referred to as *Generation*), *MCQA, and TF*. Each template is designed with appropriate few-shot exemplars and corresponding matching rules. For example, in *Generation*, due to the complexity of evaluating responses, we apply fuzzy matching techniques to compare the generated responses against predefined labels. See Appendix A.2 for further details of few-shot exemplars, fuzzy matching rules, and examples of three templates.

---

[3]We refer a recallable knowledge as the presence of at least one answer in the answer set, generated using either the greedy decoding or the temperature sampling method.

Table 2: Statistics of our benchmark dataset. We categorize whether knowledge changes over time (Time Variant) or remains constant (Time Invariant). Among five domains, we set the temporal state with dynamic (knowledge that changes within the time frame we have set) and static (knowledge that do not change within the time frame we have set). The number in parentheses represents the average change in objects per element within a dynamic dataset. See details in Appendix A.3.1.

| Time Dependency | Domain (Time Frame) | # of Relations | Structured | Format | Temporal State | # of Examples | Source |
|---|---|---|---|---|---|---|---|
| Time Variant | general (2010 - 2023) | 8 | ✓ | $(s, r, o, t)$ | dynamic (2.6) static | 8,330 8,302 | Wikidata |
| | biomedical (2020 - 2024) | 14 | ✓ | $(s, r, o, t)$ | dynamic (2.3) static | 7,345 7,345 | UMLS |
| | legal (2010 - 2023) | 6* | ✗ | QA | dynamic (1.1) static | 3,142 3,142 | CFR |
| Time Invariant | commonsense | 8 | ✓ | $(s, r, o)$ | invariant | 24,788 | CSKG |
| | math | 12 | ✓ | $(s, r, o)$ | invariant | 2,585 | Math-KG |

## 3 CHROKNOWBENCH: CONSTRUCTING A BENCHMARK DATASET

In this section, we enumerate the details of constructing **CHROKNOWBENCH**, a chronologically accumulated knowledge benchmark dataset. The CHROKNOWBENCH dataset encompasses three key aspects: time dependency (time variant and invariant), multiple domains (general, biomedical, legal, commonsense, and math), and temporal state (dynamic and static). We first categorize knowledge into two groups: knowledge that remains unchanged over time (time invariant) and knowledge that changes over time (time variant). Additionally, we classify domains based on whether they are influenced by the flow of time, considering the specificity of each domain. Finally, we categorize knowledge as either changeable (dynamic) or steady (static) within the time frame we have set.

### 3.1 TASK DEFINITION

Our primary focus is a time variant knowledge across three domains (general, biomedical, and legal) with comparisons to time invariant knowledge across two domains (commonsense and mathematics). What knowledge would be the difference between time variant and invariant? The time variant knowledge has a specific object changing across a stream of time. For example, "Cristiano Ronaldo *(s)* was a member of sports team of *(r)* Manchester United F.C. *($o_1$)* in 2009 *($t_1$)* and Real Madrid CF *($o_k$)* in 2018 *($t_k$)*". Particularly, we adopts an object-level focus, emphasizing changes in object attributes such as roles or affiliations. This approach ensures **scalability** by simplifying temporal dynamics, **precision** by capturing nuanced updates, and **flexibility** by supporting fine-grained real-world transformations without rigid relational constraints. Likewise, we identify subject and object alias for each relation, then gather yearly changed objects. After accumulating object lists $\{o_1, o_2, \ldots, o_m\}$, we de-duplicate and fill out the missing data in specific years based on available data; objects between Manchester United F.C. *($o_1$)* in 2009 *($t_1$)* and Real Madrid CF *($o_k$)* in 2018 *($t_k$)*, missing objects between 2010 *($t_2$)* to 2017 *($t_{k-1}$)* filled with existing object of 2009 *($t_1$)*. The statistic of CHROKNOWBENCH dataset is in Table 2, and detail of object-level focus is in Appendix A.3.2.

### 3.2 DATASET GENERATION

To construct dataset, we select annual knowledge sources for each domain, possible to be aligned with each elements even though the corpus does not specifically mention about time stamp. For sources with structured triplets, we identify temporal affect-able relations that typically change over time, such as *"position held"*. As time variant knowledge refers to the knowledge that has the potential to change over time, we divide it into two temporal states for more fine grained results: (1) **dynamic**, where the knowledge has evolved over the period. (2) **static**, where no change occurred during the period, though it has potential to be changed. Following the methodology outlined in Section 3.1, we track changes in objects to build the dynamic dataset, employing normalization and de-duplication for verification. Each object is checked with strict exact string match, then add into objects pool. Simultaneously, we identified unchanged objects over the same time frame to construct the static dataset. At the end, all data elements consist with an associated object pool $\{o_1, o_2, \ldots, o_m\}$ over time frames $\{t_1, t_2, \ldots, t_m\}$.

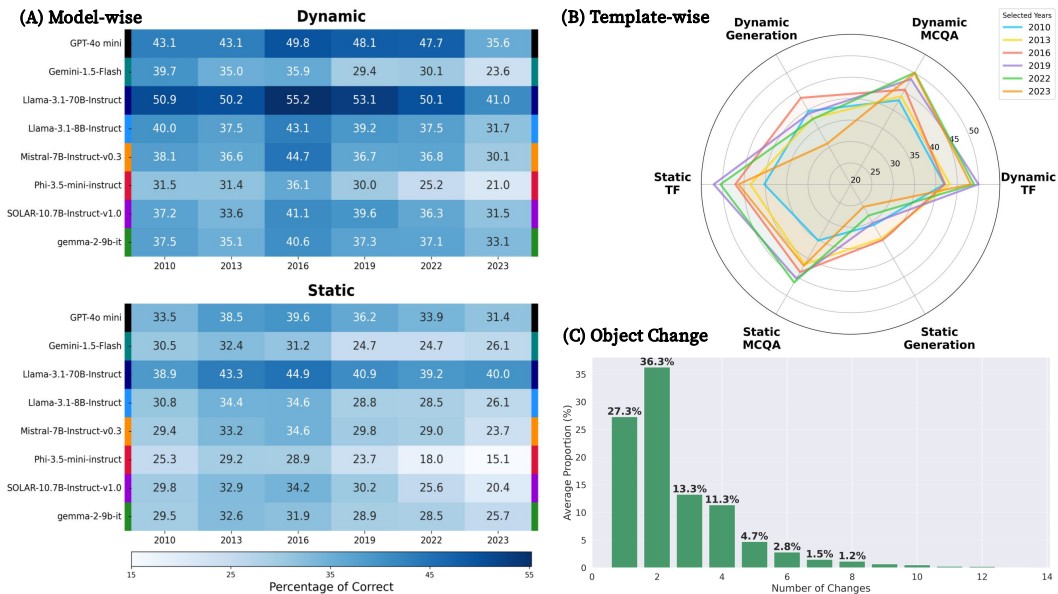

Figure 2: Performance analysis of general domain. (A) Heatmap in Generation template. For both dynamic and static datasets, a common trend across models is that performance is stronger in the intermediate years but decline recent years, reflecting the data-cutoff point. Dynamic knowledge shows more variation compared to static. Full results of total time frame is in Figure 10. (B) Template-wise performance for selected years. As time goes by, performance in generation goes low, on the other hand, MCQA and TF appeal to be rising. (C) Distribution of object changes in dynamic dataset.

### 3.3 TIME VARIANT & INVARIANT KNOWLEDGE

We sourced time variant knowledge from the general, biomedical, and legal domains. In general domain, we utilize Wikidata (Vrandečić & Krötzsch, 2014) dump to track object changes among the time frame using suggested time quantifiers. Collecting similar amounts of dynamic and static instances across eight relations, the result is formatted as $\{s, r, o, t\}$ quadruplet for each object and accompanied time stamp. For biomedical domain, we parse Unified Medical Language System (UMLS) (Bodenreider, 2004) metathesaurus data, where suggest yearly updated research in the domain, following previous work of BIOLAMA (Sung et al., 2021). Due to the slow pace of change in biomedical research, object pools in this domain shows slight expansions or contractions over time frame. In the legal domain, we employ the Code of Federal Regulations (CFR) (U.S. Government) to track regulatory changes, as they suggest collection and accumulation of change in regulations at the end of year. Starting from pre-processing unstructured xml data, we adopt a QA-like format with placeholder for object, tracked among time frame which ends to dynamic or static whether it change or not. Time invariant knowledge, which remains constant regardless of temporal context, is drawn from common-sense and mathematical domains. We process the CSKG (Ilievski et al., 2021) dataset of commonsense knowledge, and Math-KG (Wang, 2022) for covering areas like data structures and pure mathematics. Further details are provided in Appendix A.3, especially the sources and the mechanism of object-level focus. Especially, compared to TKGs (Zhang et al., 2024a), which focus on temporal snapshots, our CHROKNOWBENCH emphasizes the detailed temporal evolution of individual knowledge elements, offering better adaptability for gradual changes; see Appendix A.3.3.

## 4 EXPERIMENTAL SETUP

We enumerate the nine open-source and two proprietary LLMs: Llama-3.1-70B-Instruct and Llama-3.1-8B-Instruct (Meta, 2024), Llama-3-8B-Instruct (Dubey et al., 2024), Llama-2-7b-chat-hf (Touvron et al., 2023b), Mistral-7B-Instruct-v0.3 (Jiang et al., 2023), Phi-3.5-mini-instruct (Abdin et al., 2024), SOLAR-10.7B-Instruct-v1.0 (Kim et al., 2023), gemma-2-9b-it (Team et al., 2024b), gemma-7b-it (Team et al., 2024a) for major open-source models, and GPT-4o mini (OpenAI, 2024a), Gemini-1.5-flash (DeepMind, 2024) for proprietary models. Each model utilizes either an instruction-tuned or chat version to enhance instruction following during sampling. We focus on anlayzing trends in the chronological knowledge captured by those models, differ in corpus coverage. Details of our inference setups are in Appendix A.4.

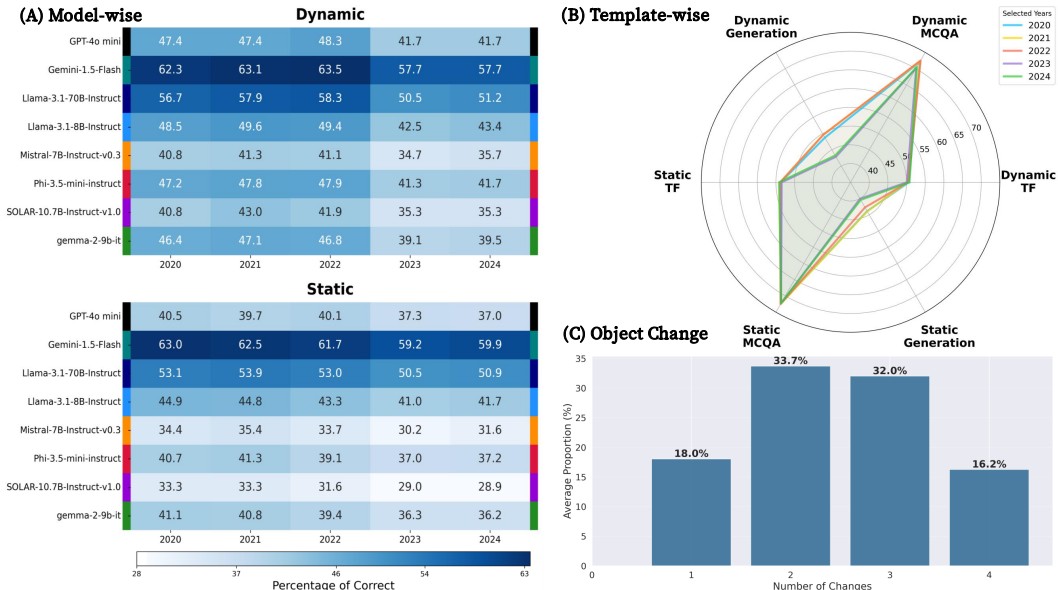

Figure 3: Performance analysis of biomedical domain. The format of figure is same as Figure 2. (A) Compared to the general domain, both dynamic and static datasets show lower variability, reflecting a domain-specific tendency toward consistency in knowledge changes. Both of them shows performance decrease between 2022 and 2023, aligning with the cutoff pattern noted in the general domain. (B) As time goes by, performance in generation declines, but MCQA and TF continue to perform well.

## 5 CHROKNOWLEDGE: CHRONOLOGICAL CATEGORIZATION OF KNOWLEDGE

In this section, we introduce **CHROKNOWLEDGE** (***Chro**nological Categorization of **Knowledge***), a sampling-based framework designed to evaluate LLMs' non-parametric chronological knowledge. Our methodology assesses the temporal capabilities of LLMs using two distinct templates, detailed in Section 2.2, and explores how current LLMs encapsulate temporal information.

### 5.1 RESULTS OF REPRESENTING KNOWLEDGE FROM LARGE LANGUAGE MODELS

For testing model's knowledge within the categorization, we sample five times for each knowledge to elicit it as possible in dynamic and static dataset. We present our findings across three different aspects: temporal-wise, template-wise, and domain-wise results. In Figure 2, 3 and 4, we depict the results for all time variant domains; general, biomedical and legal domains in main section. Time invariant datasets are presented in Appendix A.5, Figure 9.

**Temporal-wise Results.** Comparing the upper and lower panels of (A) in Figure 2, 3 and 4 provides the tendency of temporal-wise results based on the generation results. A common trend across models is a decline in performance on recent knowledge, reflecting the pretraining corpus cutoff dates. Particularly in dynamic datasets, models demonstrate strong performance on earlier knowledge but experience a steeper decline in later periods, particularly in both general and biomedical domains. In contrast, static datasets show less fluctuation, with more stable yet weaker performance, highlighting limited temporal sensitivity due to reliance on a single timestamp. These results emphasize the need for frequent model updates, especially for dynamic knowledge, to ensure temporal relevance.

**Template-wise Results.** (B) of Figure 2, 3 and 4 provide the average scores of template-wise results, for more template specificity checking in three templates: generation, MCQA, and TF. Generation templates reveal a greater decline in recent knowledge, as models rely on internal information without predefined answers. In contrast, MCQA and TF templates help models select correct answers from structured options and predefined formats, mitigating some gaps in recent knowledge. This trend is more evident in the biomedical domain with MCQA templates and the legal domain with TF, revealing how domain-specific knowledge is more effectively elicited through specific formats. Also, dynamic dataset in the general domain is more sensitive to temporal shifts than static, highlighting the importance of task-specific templates in eliciting and improving temporal robustness.

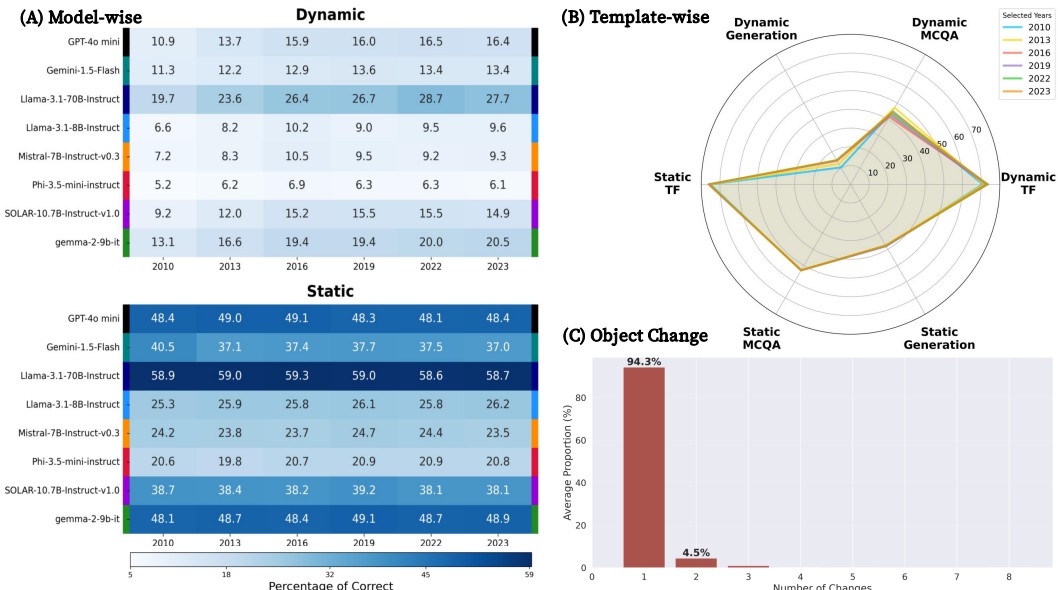

Figure 4: Performance analysis of legal domain. The format of figure is same as Figure 2. (A) Among time variant domains, legal domain shows the most stable results of static, while the gap between dynamic and static datasets is the largest among domains. (B) When it comes to each template, generation shows the lowest performance, while TF settings perform extraordinarily well in answering correctly. (C) In the legal domain, the distribution shows the lowest number of object changes over time, supporting the conclusion of the stable results in the heatmap.

**Domain-wise Results.** Comparing the Figure 2, 3 and 4 provide the tendency of domain-wise results, demonstrating distinct domain-specific characteristics of temporal knowledge change. In general domain, the models show a decline in recent knowledge, with a more unstable distribution of scores, which stems from domain's nature; changes in relations *'position held'* or *'member of sports team'* are more sensitive to temporal cues, leading to higher variability. Here, MCQA setting offers some resilience as it mitigates the knowledge decline observed in generation templates across dynamic and static datasets over different years. Every domain shows similar distribution of object changes with benchmark statistics, comparing (C) in each figure with Figure 8. This indicates that the models perform robustly as intended by the benchmark, without bias toward specific low changed objects. We provide more details for legal domain and time-invariant knowledge in Appendix A.5.

Overall, domain-specific characteristics significantly influence LLMs' temporal knowledge representation. More stable domains like biomedical and legal exhibit consistent performance with time invariant knowledge, while general domain shows more inconsistency. These insights underscore the need for tailored strategies to enhance LLMs' temporal knowledge capabilities.

# 6 CHROKNOWPROMPT: CHRONOLOGICAL KNOWLEDGE PROMPTING

## 6.1 CHRONOLOGICAL CATEGORIZATION

In previous section, we demonstrate whether open-source and proprietary models possess specific knowledge at various time stamps. However, this does not sufficiently assess the models' understanding of knowledge within a chronological progression. As Zhao et al. (2024) suggest, knowledge influenced heavily by temporal factors as general domain can still vary in more stable situation like static dataset. To address this, we first reclassify the models' responses using a refined categorization scheme, allowing for a more comprehensive evaluation of temporal knowledge across all years.

Figure 6 illustrates how it works: (1) **Known** for the precise temporal alignment if the model correctly identifies all relevant objects for a given knowledge category at each specific year; (2) If model fails to match the correct objects for every year, we refer it as **Unknown** for indicating incomplete or misaligned temporal knowledge; (3) The model accurately responds either just before or after a specific year but fails for others, signifying outdated information or forgotten legacy knowledge due to continuous updates (**Cut-off**); (4) The model correctly identifies some objects for a given year but not others, reflecting an incomplete understanding of the temporal knowledge (**Partial Known**).

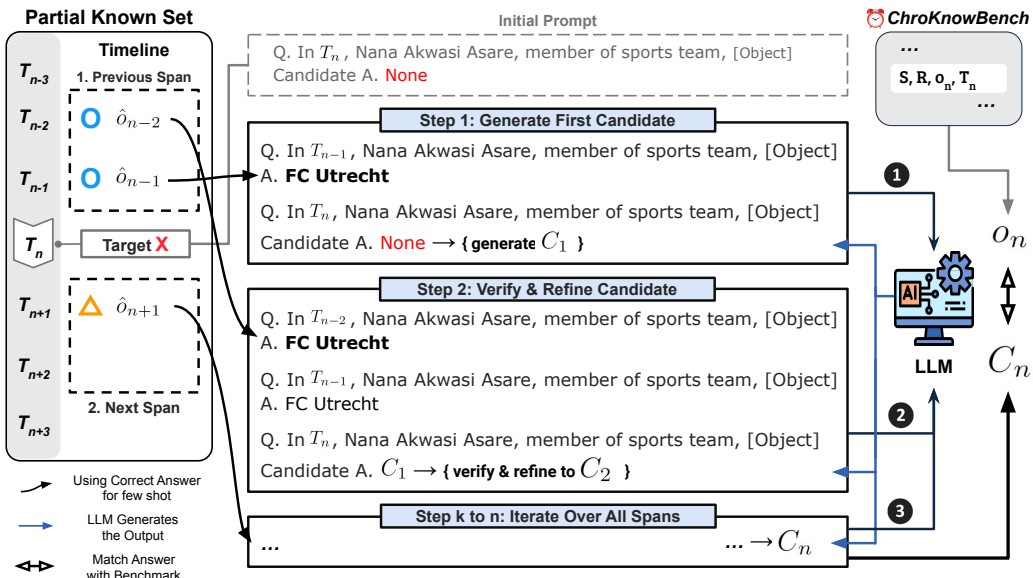

Figure 5: Overview of **ChroKnowPrompt**. The algorithm systematically traverses step by step, appending each span's correct answer as few shot for each steps. The range of each previous and next span is predefined, with the order of nearest time stamp from target $T_n$. The model suggests last candidate answer $C_n$, verified an d refined through several steps, which ends to be checked with the object $o_n$ in original **ChroKnowBench**.

Our main focus is on the **Partial Known** category, where models demonstrate substantial temporal knowledge but fail to answer correctly for all years, often showing confusion between correct answers. For example, Nana Akwasi Asare *(s)* was a member of sports team of *(r)* FC Utrecht *($o_n$)* in 2011 *($t_n$)*, but the model incorrectly identifies the team as FC Groningen, despite answering correctly with FC Utrecht for 2010 *($t_{n-1}$)* and in 2012 *($t_{n+1}$)*. At this point, we hypothesize that when the model gets one time stamp wrong, a more explicit focus on the temporal aspects surrounding that time span could help it generate more accurate answers. This is the core idea behind **CHRO-KNOWPROMPT**.

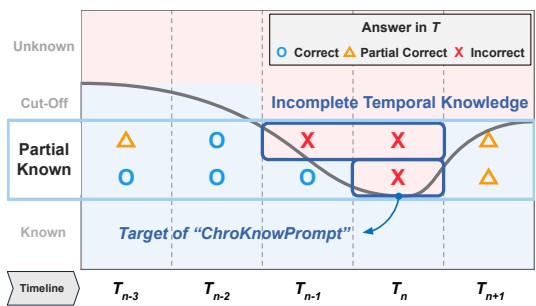

Figure 6: Chronological categorization based on each answer with its time stamp. If the model answer correctly for all, it is re-categorized as **Known**. The target of **ChroKnowPrompt** is **Partial Known**, which confuses its knowledge among the whole time stamps.

## 6.2 METHOD

We introduce a chronological prompting technique for non-parametric method to elicit chronological knowledge, aimed at bridging knowledge gaps by utilizing multiple temporal snapshots. This method enhances the model's reasoning by systematically integrating knowledge from different time stamps, enabling in-depth traverse. Our method is inspired by non-parametric editing techniques, such as Zhong et al. (2023) and Zheng et al. (2023), described in detail in Appendix A.1.

Figure 5 illustrates an example of the chronological prompting process. From a target year $t_n$, the algorithm systematically traverses the preceding years $(t_{n-1}, t_{n-2}, \ldots)$ in the 'Previous Span' and the subsequent years $(t_{n+1}, t_{n+2}, \ldots)$ in the 'Next Span'. For each traversed year, the most representative object $\hat{o}_k$ is selected from the *Correct* (represented by circle) and *Partial Correct* (represented by triangle) categories in Table 1 using majority voting.

Starting with the initial prompt containing the target time $t_n$, subject $s_n$, and relation $r_n$, the nearest year in the previous span is appended above the initial prompt with the selected object, forming the first step. The model then generates a candidate answer $C_1$ for $t_n$ using this augmented prompt. Next, our CHROKNOWPROMPT iteratively adds prompts from progressively earlier years, refining

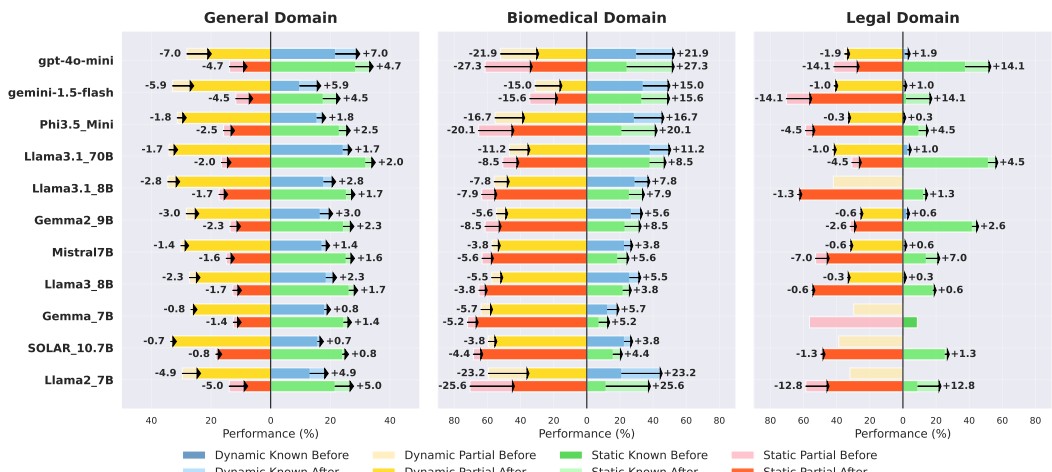

Figure 7: Results of **ChroKnowPrompt** across multiple domains with unchanged objects. For each domain, the left space represents the percentage of **Partial Known**, and the right represents the percentage of **Known**. Each model includes results for both dynamic (yellow-blue bar) and static (red-green) datasets, with arrows indicating the actual increase. As shown in plots, the most effective results are observed in the biomedical domain, where the unchangeable characteristic is stronger than the general domain. While the static dataset of the legal domain shows improvement, many models struggle with unstructured format, resulting in the lowest performance among the dynamic dataset.

the candidate answer by verifying its consistency across contexts (from $C_1$ to $C_2$). This backward traversal continues until a predefined span range is reached. Once the previous span is completed, our algorithm performs forward traversal by appending objects from subsequent years below the target year, further generating and verifying candidate answers. If there are no previous or next years available, the process proceeds on only one side.

Upon completing all traversals, the final candidate answer $C_n$ is compared against the benchmark object for $t_n$. If the candidate answer aligns correctly with the object for $t_n$, appropriately reflecting the temporal contexts, the knowledge categorization for $t_n$ is updated to *Chrono-Correct*, which is equivalent to *Correct* for chronological assessments. In Appendix A.7, we provide the detailed steps.

# 7 EXPERIMENTAL RESULTS & ANALYSIS

## 7.1 RESULTS OF CHROKNOWPROMPT

Details of task configuration is in Appendix A.8. Figure 7 presents the effect of chronological prompting on unchanged objects across different models. Results show the rise of percentage in *Known* category with decreasing *Partial Known*, indicating the increase by *Chrono-correct*. Significant improvements are observed in the biomedical domain (average increase of 11.5%), while general and legal domains show smaller gains (2.8% and 3.1%, respectively). Both proprietary models (GPT4o-mini, Gemini-1.5-flash) and the massive open-source model (Llama-3.1-70B-Instruct) perform well, while smaller open-source models like Llama-2-7B-chat-hf also show notable improvements despite being outdated. This indicates that chronological prompting effectively enhances knowledge recall without requiring external retrieval systems. But in the legal domain, models struggle to recall knowledge due to the complexity of unstructured, context-rich data, especially for dynamic dataset.

## 7.2 ANALYSIS

**Results in changed objects** Table 5 compares results for changed and unchanged objects in dynamic datasets across domains. As shown in Section 7.1, **ChroKnowPrompt** effectively aids models in recalling unchanged objects in both dynamic and static datasets. However, its performance on changed objects in dynamic datasets remains limited, achieving only 10–30% performance of unchanged object cases (an average of 0.4 in general domains). This highlights, despite the helpfulness and applicability of **ChroKnowPrompt** in addressing the chronological gaps of knowledge, recalling all historical changes of objects using prompting alone remains an exceptionally challenging problem, particularly in complex contexts such as the legal domain. These findings emphasize the need for further research into effective parametric editing techniques to assist temporal knowledge handling.

**Effects of chronological span**   To elucidate the mechanisms of chronological prompting, we analyze the impact of incorporating the next span in chronological contexts. As shown in Table 6, the total span (both previous and next) yields higher scores than using only the previous span with the degree of improvement varying by domain. In the biomedical domain, the total span nearly doubles the score of the previous span alone (12.0 vs. 6.7), while the general domain shows a modest increase (2.8 vs. 1.8). Model-specific temporal sensitivity also varies: Llama-2-7b-chat-hf effectively utilizes next spans, whereas Gemini-1.5-flash and SOLAR-10.7B-Instruct-v1.0 benefit more from previous spans. These suggest that differences in temporal context utilization and the coverage of pretraining corpus may influence models' sensitivity and knowledge recall across time frames.

**Effects of chat prompting**   Additionally, we analyze various chat models (before instruction-tuning), including Llama-2-7b-chat-hf, as chronological prompting may enhance both current and legacy models by leveraging temporal context. We evaluate three open-source chat models: mpt-7b-chat (Research, 2023), Pythia-Chat-Base-7B (Biderman et al., 2023), and nemotron-3-8b-chat-4k-sft-hf (Zhang et al., 2023a). As shown in Table 6 and 7, **ChroKnowPrompt** is not particularly effective for chat models. Only mpt-7b-chat achieves a comparable peformance to Llama-2-7b-chat-hf (an average increase of 11.4), while Pythia-Chat-Base-7B shows almost no improvement.

## 8   RELATED WORK

Since the emergence of LMs, deriving knowledge from language model is extensively studied, such as probing tasks (Hewitt & Manning, 2019), LAMA (Petroni et al., 2019) and BioLAMA (Sung et al., 2021). Then, many subsequent studies follows to exploit, (1) how LLMs define knowledge (Yu et al., 2023; Zhang et al., 2023b; Gottesman & Geva, 2024), (2) how these models represent it (Chen et al., 2024a;b; Wang et al., 2024d), and (3) how manipulate misleading part (Wang et al., 2023; Gutiérrez et al., 2024; Wu et al., 2024a). Based on them, recent investigations of knowledge highlight the dynamic nature of evolving facts and suggests that contradictions within the training data may lead to knowledge conflicts (Marjanović et al., 2024; Chang et al., 2024; Wang et al., 2024a; Xu et al., 2024; Jin et al., 2024). And knowledge overshadowing (Zhang et al., 2024c) reveals phenomena where certain conditions overshadow other facts, leading to misleading information (i.e., hallucinations).

In other view point, exploring temporal knowledge starts from using Wikidata, a static format of knowledge in triplet: subject, relation, and object, originated from extracting literature-based knowledge (Hahn-Powell et al., 2017). Following pioneers like TimeQA (Chen et al., 2021) and TemporalWiki (Jang et al., 2022a), many works dealing with temporal and continuous knowledge flow (Zhang & Choi, 2021; Dhingra et al., 2022; Jang et al., 2022b; Liska et al., 2022; Nylund et al., 2023; Zhu et al., 2023; Khodja et al., 2024; Zhang et al., 2024d) consist in line of it. Building upon their achievement, CarpeDiem (Kim et al., 2024) emerges to simply identify whether knowledge is outdated or not, and DyKnow (Mousavi et al., 2024) maps various models' knowledge distribution. Also, (Zhao et al., 2024) makes dramatic work: align model into one fixed age. Though those impressive works, we try to broad the coverage of temporal knowledge. We utilize various templates to elicit the knowledge of LLMs, broaden the coverage of time stamps, and differentiate domains that should change based on a temporal perspective with those that should remain constant.

## 9   CONCLUSION, LIMITATION, AND FUTURE WORK

Overall, our work highlights the critical role of temporal context in knowledge evaluation and introduces a framework for improving the temporal capabilities of future language models. We present **CHROKNOWBENCH**, a benchmark for assessing temporal knowledge across diverse domains, and our **CHROKNOWLEDGE** framework, which evaluates LLMs' chronological knowledge through three types of templates. Our findings indicate that while models often recall facts with time stamps, they struggle with capturing full temporal boundaries, with reliance on rigid formats like MCQA and TF. By using CHROKNOWPROMPT, we improve knowledge recall by reducing ambiguous *Partial Known* and increasing *Known*, particularly in biomedical domains with strong performance on unchanged objects in both proprietary and open-source models. However, challenges persist in dynamic datasets and in unstructured, context-rich formats, which amplify the difficulty of capturing temporal evolution solely with prompting. Future work will focus on the need for parametric techniques to complement prompting, enabling better alignment with changing objects and complex temporal dependencies to enhance LLMs' temporal accuracy across various domains.

ACKNOWLEDGMENTS

This work was supported in part by the National Research Foundation of Korea [NRF-2023R1A2C3004176, RS-2023-00262002], the Ministry of Health & Welfare, Republic of Korea [HR20C002103], and the ICT Creative Consilience program through the Institute of Information & Communications Technology Planning & Evaluation (IITP) grant funded by the MSIT [IITP-2025-2020-0-01819].

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

# A APPENDIX

## A.1 SUPPLEMENTARY STUDIES IN KNOWLEDGE EDITING

### A.1.1 PARAMETRIC KNOWLEDGE UPDATE

Considering update of LLMs are in two types, parametric and non-parametric (Wang et al., 2024c), a classical way of parametric update is using fine-tuning (Ghosal et al., 2024; Mecklenburg et al., 2024; Ge et al., 2024). While the method extends to LoRA (Hu et al., 2022), QLoRA (Dettmers et al., 2023), and Melo (Yu et al., 2024), as well as continual learning approaches such as GRACE (Hartvigsen et al., 2024) and WISE (Wang et al., 2024b), the parameter accessibility of open-source LLMs like

the Llama series (Touvron et al., 2023a) enables techniques such as MEND (Mitchell et al., 2022a), ROME (Meng et al., 2022), and MEMIT (Meng et al., 2023) to emerge. Those local editable methods are effective, and still try to improve specificity and generalizability.

### A.1.2 Non-parametric Knowledge Update

In contrast, for black-box LLMs, updates rely on non-parametric knowledge methods (Onoe et al., 2023), such as SERAC (Mitchell et al., 2022b), MeLLo (Zhong et al., 2023), and IKE (Zheng et al., 2023). They align with two key trends: (1) Mitigating catastrophic forgetting, where the model loses previous knowledge, by not directly updating parameters. (2) Exploiting abilities of prominent black-box LLMs like GPT-o1 (OpenAI, 2024b) and Gemini (Team et al., 2023), as we cannot access to parameters. Another concern in knowledge update is they often focus only structured format, pointed out that current methods struggle to update unstrctured data effectively (Wu et al., 2024b). In this paper, we focus on non-parametric knowledge updates accomodating a broad range of input formats (structured and unstructured) to represent knowledge across diverse domains, depending on the use of various white-box and black-box LLMs.

### A.2 Details of Eliciting Knowledge: Few-shot Exemplars, Fuzzy Matching Rules, and Examples of Three Templates

### A.2.1 Few-shot Exemplars

To obtain the few-shot exemplar pool $D$, we leverage additional data collected using the same process as in CHROKNOWBENCH. Specifically, for each individual relation type, We gather four exemplars for the general domain, and eight for the biomedical and legal domains. We then generate the few-shot exemplar set $D_i$ by sampling four exemplars from $D$, which serves as actual demonstrations within a prompt. This process is repeated for every timestamp to ensure comprehensive temporal coverage.

### A.2.2 Fuzzy Matching

We utilize the `rapidfuzz` library to compare the model's responses with the predefined labels. As the model's answer may a little bit different with complicated objects in specialized domains, such as the difference in order of words or upper and lower cases, using fuzzy match enables more rapid but still reliable quality without facilitating external NLI mechanisms.

Specifically, we employ a `token_set_ratio` metric with a threshold value set to 70 to determine a match. `token_set_ratio` is a metric used for comparing the similarity of two strings in a flexible manner, extending the functionality of the `token_sort_ratio`. In the preprocessing stage, the strings undergo tokenization, removal of punctuation, and conversion to lowercase. The tokens are then sorted in alphanumeric order before the similarity ratio is computed. This makes it useful for comparing strings where the word order may differ but the content is similar.

The key distinction of `token_set_ratio` lies in its incorporation of set operations, where duplicate words are removed. After eliminating repeated tokens, the same preprocessing steps as in `token_sort_ratio` are applied. When performing the comparison, the method checks if all tokens from the shorter string are contained within the longer string, making the approach particularly suited for cases where one string is a subset of the other. This flexible matching often results in higher accuracy for comparing strings with similar content but different structures, as illustrated by the example where a score of 100 is achieved when all tokens from the second string are present in the first.

### A.2.3 Examples of Three Templates

We provide three templates of *Generation*, *MCQA*, and *TF* in the end of the Appendix for the better readability. For example, in Table 8 and Table 9, our target year is 2020 ($t$) to generate answer candidate of position held ($r$) by Donald Tusk ($s$).

### A.2.4 ITERATIVE DISTRACTOR GENERATION

For the Commonsense dataset, the objects corresponding to a given subject and relation are often ambiguous. When constructing compelling distractors, there is a higher likelihood (about 20%) of creating options that are actually correct answers rather than intended incorrect ones, compared to other datasets. Therefore, we include an additional verification process after generating the distractors, as outlined in Algorithm 1. Specifically, we formulate multiple-choice questions using the problem and the generated distractors, then use GPT-4o to select all correct answers. If it identifies more than one correct answer, we refine the distractors based on a prompt to recreate incorrect options.

---

**Algorithm 1:** Iterative Distractor Generation Algorithm

**Data:** Subject $s$, Relation $r$, Set of correct objects $\mathcal{O}_{\text{correct}}$
**Result:** Refined multiple-choice question $q$

1 Initialize conversation history $\mathcal{H} \leftarrow \emptyset$
2 Initialize number of selected options $n \leftarrow 1$
3 **while** $n > 0$ **do**
4     $\mathcal{D} \leftarrow \text{LLMResponse}(s, r, \mathcal{H})$   // Generate three incorrect options
5     $q \leftarrow \text{ComposeQuestion}(s, r, \mathcal{D})$   // Compose question using the generated distractors
6     Append $q$ to $\mathcal{H}$
7     $\dashv \leftarrow \text{LLMResponse}(q)$   // LLM generates a response by solving the question
8     $\mathcal{S} \leftarrow \text{LLMResponse}(\dashv)$   // Extract selected options using LLM
9     $n \leftarrow |\mathcal{S}|$   // Number of options selected by LLM
10     **if** $n > 0$ **then**
11         $p \leftarrow \text{CreatePrompt}(s, r)$   // Create another prompt for regenerating distractors
12         Append $p$ to $\mathcal{H}$   // Add regeneration prompt to the conversation history
13         $\mathcal{D}_{\text{new}} \leftarrow \text{LLMResponse}(s, r, \mathcal{H})$   // Generate new set of distractors
14         $\mathcal{D}[\mathcal{S}] \leftarrow \mathcal{D}_{\text{new}}[1:n]$   // Replace selected options
15 **return** $q$

---

### A.3 DETAILS OF BENCHMARK DATASET

### A.3.1 STATISTICS OF OBJECT CHANGES IN DYNAMIC DATASET

The statistics of object changes among dynamic dataset in three time variant domains are in Figure 8. The figure shows how many objects have been changed among the time frame of each elements, and its distribution is proposed with the percentage of that elements across total number of dataset. The average number of object changes is 2.6 and 2.3 for general and biomedical dynamic dataset, while most of the element in legal domain has only one object changes among time frame. The biomedical domain shows the least skewness with a balanced cumulative distribution of changes, unlike the general domain, which is moderately skewed with broader change frequencies. The legal domain is highly skewed, with most changes concentrated in a single occurrence, lacking cumulative progression.

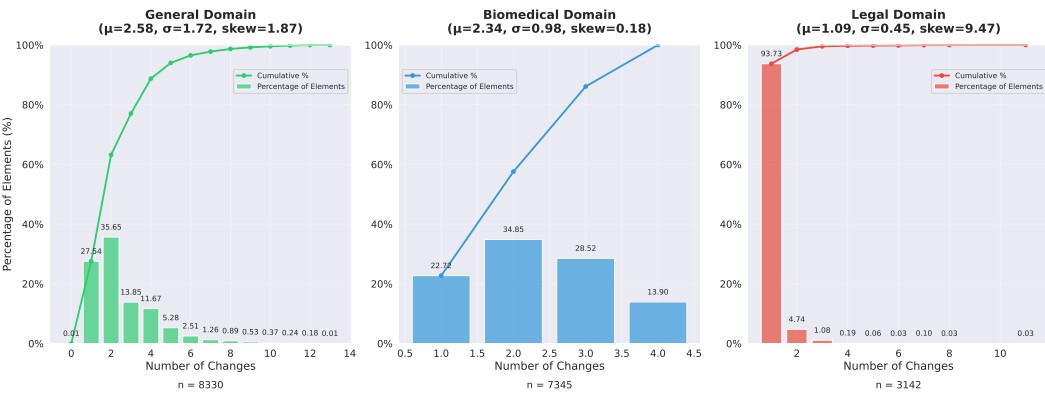

Figure 8: Statistics of object changes among dynamic dataset in three time variant domains.

### A.3.2 OBJECT-LEVEL FOCUS IN BENCHMARK GENERATION

Our CHROKNOWBENCH emphasizes object-level changes as the core metric for assessing temporal knowledge dynamics. This approach reflects a deliberate balance between scalability, precision, and interpretability, aligning with the methodological goals of the benchmark.

**Design and Scope**   The benchmark evaluates how well models can align and reason about temporal knowledge by focusing on object-level transformations at specific time points. For instance: Query 1: *"In 2001, Zidane was a player at Real Madrid."* and Query 2: *"In 2010, Zidane was a coach."* By treating these as distinct evaluations, CHROKNOWBENCH isolates object-level transitions—such as roles or affiliations—while keeping the subject and relation fixed. This design ensures a structured and scalable evaluation process that captures significant, interpretable changes in temporal knowledge.

**Reason of focusing on Object-Level Changes**

- **Scalability:** The object-level focus simplifies the complexity of tracking temporal dynamics in subject-relation-object triples. By avoiding the combinatorial challenges associated with relational changes, this approach ensures clear and scalable evaluations.
- **Precision:** Object-level changes, such as shifts in roles or affiliations, capture more nuanced updates compared to relational changes, which often default to binary states (e.g., "is a player" → "is not a player"). This granularity enhances the depth of the evaluation.
- **Flexibility:** Unlike traditional Temporal Knowledge Graphs, which may impose rigid relational structures, the object-centered approach accommodates fine-grained changes in knowledge. This flexibility enables precise and interpretable assessments, particularly for dynamic, real-world transformations.

By centering evaluations on object-level changes, CHROKNOWBENCH delivers a robust framework for measuring temporal knowledge dynamics, balancing methodological rigor with practical scalability.

### A.3.3 COMPARISON WITH TEMPORAL KNOWLEDGE GRAPHS

Temporal Knowledge Graphs (TKGs) are one of the well known approach for addressing temporal knowledge (Jung et al., 2020; Li et al., 2021; Zhang et al., 2024a). Our CHROKNOWBENCH also aim to model time-sensitive knowledge, but we adopt different approaches to structuring and interpreting temporal information. In this section, we compare these two paradigms across several dimensions, with particular attention to their handling of temporal snapshots, knowledge dynamics, and suitability for domains such as biomedical data.

**Temporal Snapshot vs. Knowledge-Centric Tracking**   TKGs organize events based on temporal snapshots, incorporating multiple events occurring within the same timestamp into a single graph representation. This approach emphasizes the relationships between events at a specific time point, making it highly suitable for scenarios where the context of concurrent events is critical (e.g., *(North America, Host a visit, Business_Africa, 2010)*, *(Barack Obama, Consult, North America, 2010)*) (Zhang et al., 2024a). By connecting these events, TKGs enable reasoning about their inter-dependencies and broader temporal patterns.

In contrast, CHROKNOWBENCH adopts a knowledge-centric perspective, focusing on the temporal evolution of individual knowledge elements $(s, r, o)$ over time. Instead of aggregating multiple events into a single snapshot, CHROKNOWBENCH tracks changes in object $(o)$ values for each relation $(r)$ across a timeline like the example in Section 3.1 This approach highlights the evolution of specific knowledge and ensures comprehensive tracking of its temporal progression.

**Handling Specific Domains**   A key limitation of TKGs lies in their reliance on well-defined temporal snapshots. While this approach is effective for aggregating and reasoning about concurrent events, it becomes less suitable for domains like biomedical data, where changes often unfold gradually over time and are not tied to distinct temporal snapshots. For instance, the development of a medical treatment over a decade may involve incremental advancements that cannot be neatly encapsulated within discrete, event-based snapshots.

Table 3: Comparison of TKGs and CHROKNOWBENCH based on their temporal modeling approaches. TKGs focus on temporal snapshots aggregating multiple events, while CHROKNOWBENCH emphasizes tracking the temporal evolution of individual knowledge elements. This table highlights their strengths, weaknesses, and domain applicability.

| Aspect | TKGs | CHROKNOWBENCH |
|---|---|---|
| **Temporal Focus** | Temporal snapshots aggregating multiple events | Temporal evolution of individual knowledge elements |
| **Domain Applicability** | Suitable for well-defined, event-rich domains (e.g., geopolitical, social networks) | Applicable not only temporal, but also gradual or implicit changes (e.g., biomedical, legal) |
| **Handling of Gradual Changes** | Limited by snapshot granularity | Effective through continuous tracking |
| **Limitations** | May overlook fine-grained changes in individual knowledge | May overlook broader event interdependencies |

CHROKNOWBENCH overcomes this limitation not merely by dividing time into yearly intervals, but by prioritizing the temporal progression of individual knowledge elements. This distinction lies in its focus on tracking and organizing the dynamic and static changes of specific objects over time. While TKGs aggregate multiple concurrent events within a single snapshot, CHROKNOWBENCH constructs yearly object pools that capture the fine-grained evolution of a specific knowledge element across its temporal trajectory. These object pools allow CHROKNOWBENCH to explicitly track updates and fill gaps in data, ensuring a cohesive and complete representation of knowledge.

Furthermore, CHROKNOWBENCH incorporates the concept of dynamic and static datasets, categorizing knowledge based on its temporal variability. This approach enables detailed modeling of knowledge that evolves gradually while preserving distinctions from knowledge that remains unchanged over time. By avoiding the rigid aggregation of unrelated events and instead focusing on the chronological development of individual elements, CHROKNOWBENCH provides a more precise framework for fine-grained analysis in those specialized domains.

**Comparative Summary**   TKGs excel at modeling inter-event relationships within temporal snapshots, making them effective for domains where concurrent event dependencies are critical, such as geopolitical analysis. However, they struggle with gradual changes and unstructured data, limiting their applicability in multiple domains. In contrast, CHROKNOWBENCH focuses on the detailed temporal evolution of individual knowledge units, leveraging object pools to capture gradual changes effectively, particularly in specialized domains like biomedical, and legal regulations.

Table 3 provides a comparative overview of the two paradigms.

### A.3.4   SOURCE AND APPROACH OF BIOMEDICAL DOMAIN

In the biomedical domain, we follow previous work of BIOLAMA (Sung et al., 2021) framework to parse Unified Medical Language System (UMLS) yearly metathesaurus data. In the range of 2020 to 2024, we gather instances in 14 relations, resulting 7k for each dynamic and static dataset. Here, by considering domain specificity that the slow pace of change typical of long-term research, the object pool is slightly expanded or narrowed in that period; *Autonomic nerve structure*, with the relation *has indirect procedure site*, has a slightly broader scope in 2024, including additional objects like *Neurolytic autonomic nerve block* alongside previous objects such as *Intravenous regional autonomic block*. The format is same with general domain, $\{s, r, o, t\}$ quadruplet.

### A.3.5   SOURCE AND APPROACH OF LEGAL DOMAIN

In the legal domain, we create a benchmark dataset based on the Code of Federal Regulations (CFR) from 2010 to 2023. We first extract paragraph-level data from regulatory documents for each year and employ Python's difflib library to detect changes between paragraphs across adjacent years (e.g., 2011 to 2012). Careful filtering is applied to ensure that only paragraphs with minor modifications (e.g., single-word updates or subtle phrasing changes) are retained.

To further analyze the dataset, we utilize the spaCy en_core_web_lg model to detect named entities in the paragraphs and assess whether these changes involve modifications to the detected entities.

Despite noise introduced by the NER model, we initially identify around 56K changes for near-year comparisons. These changes are grouped into sequences of years to track alterations over time, while filtering out paragraphs that are introduced or removed in intermediate years. Ultimately, we focus on paragraphs present in all years between 2010 and 2023, resulting in 8,793 paragraphs.

We then apply GPT-4o-mini to assess whether the detected changes are semantically meaningful, excluding minor corrections like typographical fixes or abbreviations. This results in a refined set of 4,362 meaningful updates. Additionally, we select 4,746 unchanged paragraphs containing entities detected by the NER model. For each paragraph, we format the changes as fill-in-the-blank tasks, where the modified part is replaced with a blank, providing a rich resource for studying legal text evolution over time.

### A.3.6 Source and Approach of Commonsense and mathematics

In the commonsense domain, we utilized the CSKG dataset presented in the CSKG paper. Unlike the BIO dataset, the object lists for each triplet in this dataset consist of synonymous terms, allowing multiple triplets to share the same subject and relation. In such cases, the objects appearing in each triplet carry distinct meanings. Out of the 6 million triplets, we merged the objects of triplets that have the same subject and relation into a single set, and then sampled x number of triplets from this collection.

In the mathematics and data structure/algorithm domain, we utilized the Math-KG dataset introduced in the Math-KG paper. This dataset, originally in Chinese, stores multiple objects with the same subject and relation across different triplets. Each object was translated into English using GPT-4, after which the objects from triplets sharing the same subject and relation were merged to construct a final dataset consisting of 22k triplets.

For the CommonSense dataset, the answers (objects) corresponding to a given subject and relation are often ambiguous. Consequently, when constructing compelling distractors, there is a higher likelihood (about 20%) of creating options that are actually correct answers rather than intended incorrect ones, compared to other datasets. Therefore, we include an additional verification process after generating the distractors, as outlined in Algorithm 2. Specifically, we formulate multiple-choice questions using the problem and the generated distractors, then ask GPT-4o to select all correct answers. If it identifies more than one correct answer, we refine the distractors based on a prompt to recreate incorrect options.

### A.4 Inference Setting

We evaluates all models using vLLM (Kwon et al., 2023) system, supporting features of efficient KV cache memory management that dramatically decrease inference time. All white box LM inference is conducted by vLLM with hyper-parameter: BFloat16, fixed seed, two kinds of temperature based on each sampling setting(greedy decoding with 0.0, and high temperature with 0.7). The precision is done with eight NVIDIA A100 GPUs(80GB).

### A.5 Details of ChroKnowledge in Legal and Time-Invariant Domain

Figure 4 shows the result of legal domain. Among time variant domains, legal domain shows the most stable results of static, also minimal decline in dynamic dataset. This indicates the domain specificity, which has less frequent yearly changes (almost cases has one change of object in total time frame) and change continues across many time stamps. Also, model's capability for specific task setting is influential in legal domain like the result of MCQA and TF shows, where gap between generation and other templates are many times larger than in other domains.

For commonsense and mathematics in Figure 9, arbitrary years based on the biomedical domain were used, from 2020 to 2024. The left side of result shows the tendency of generation templates, and the middle side is the tendency of MCQA templates, and the last one for TF templates. Each results measure the percentage of *Correct* answer, represented as line plots. Results show minimal variation, aligning with the stable nature of these knowledge types. This consistency confirms that time-invariant knowledge is well-preserved across models. For template wise comparison, generation cases show a way little gap between models, while MCQA tasks show the different between models,

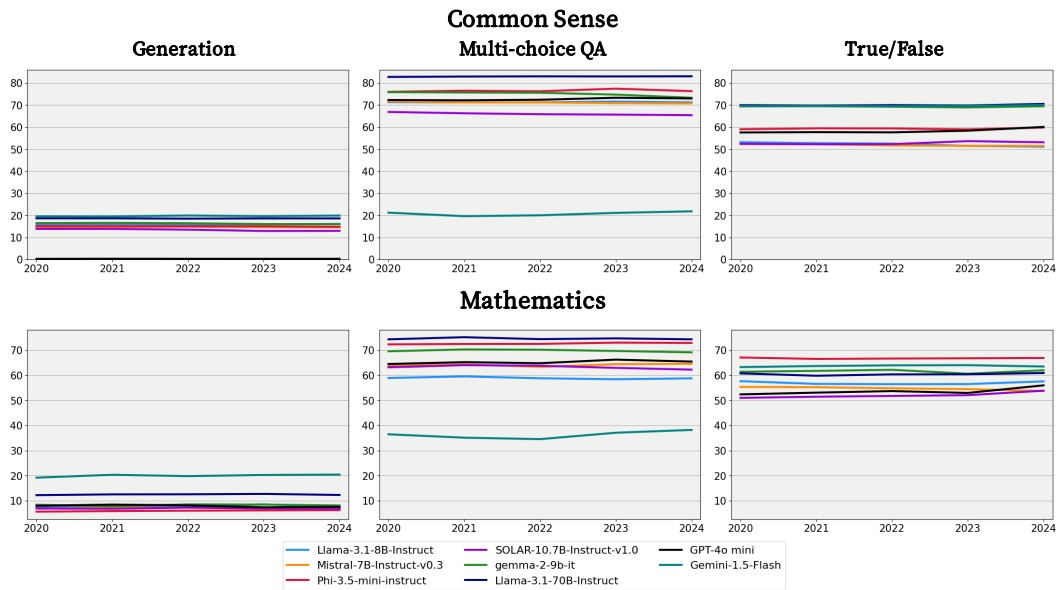

Figure 9: Performance analysis of common-sense and mathematics domains. Three line plots represent each template's results: Generation, MCQA and TF. All model shows clearly the domain specific characteristics, which is invariant knowledge even it comes with temporal attributes. Overall results are lower in generation templates, as it is challenging for models to correctly recall exactly one object in these domains (e.g., 'subject': 'Parent', 'relation': 'Synonym' has more objects later than 'Ancestor')

which is aligned with the findings from other time variant domains: the ability of each model's specialized task affects its knowledge recall ability.

About the overall performance quality, the result of time invariant shows lower performance as the models generate one object per each knowledge, while the time invariant knowledge's coverage is wider than other domains. This tendency is alleviated by using MCQA and TF templates, which ends of the rationale for helpfulness of using multiple templates to check knowledge.

## A.6    TOTAL TIME FRAME RESULT OF CHROKNOWLEDGE

Figure 10–12 represent the total results with total time frames in general, biomedical and legal domain, including chat template models like mpt-7B. Figure 13– 17 represents the total results of template-wise performance in **ChroKnowledge**. Each domain's result is separated into two temporal state: Dynamic and Static. Every spider plots consist with three template: Generation, Multi-choice QA, and True/False. Each statistics refer the percentage of *Correct* answer, same as Figure 2.

## A.7    ALGORITHM OF CHROKNOWPROMPT

The overall scheme of ChroKnowPrompt is down below. As described in Section 6.2, the algorithm starts from making initial prompt with target time $t_n$, subject $s_n$ and relation $r_n$ from target triplet. As the initialized candidate answer is None that model cannot properly answer for that target year, the algorithm also starts with make a empty list of candidate answer list $A$. and accumulated prompt $\mathcal{P}$. Then, the algorithm checks the correct object within each time span $P$ and $N$. If one of those span has no correct object, the algorithm passes that side of traversal. It the preparation is all done, the first step in previous span (if no previous span exists, the nearest next span) begins with selecting object $\hat{o}$ by majority voting. Appending prompts in each step, the model is asked to generate or verify and refine the answer $C$ of each step, like in Figure 5. After all step is done, the last candidate answer, which is the most refined result, is being checked with the original target object $o_n$ coming from target triplet. If it is matched (we also used fuzzy match in here), the category of *Incorrect* is updated to *Chrono-correct*.

---

**Algorithm 2:** Chronological Prompting Algorithm

---

**Data:** Correct set $\mathcal{C} = \{(t_i, c_i)\}$, target time $t$, triplet $(s, r, o)$, Prev span $P$, Next span $N$
**Result:** List of candidate answers $\mathcal{A}$, Updated Category

1  Initialize accumulated prompt $\mathcal{P} \leftarrow \emptyset$
2  Initialize candidate answer $a \leftarrow \emptyset$
3  Initialize candidate answer list $\mathcal{A} \leftarrow \emptyset$
4  $\mathcal{T}_{\text{prev}} \leftarrow$ time before $t$ in $\mathcal{C}$ up to span $P$    // Find correct object in previous time
5  $\mathcal{T}_{\text{next}} \leftarrow$ time after $t$ in $\mathcal{C}$ up to span $N$    // Find correct object in next time
6  **if** $\mathcal{T}_{prev} = \emptyset$ **then**
7     Skip backward traversal and process next years only

8  **if** $\mathcal{T}_{next} = \emptyset$ **then**
9     Skip forward traversal and process previous years only

10  **for** $t_p \in \mathcal{T}_{prev}$ **do**            // Process previous years first
11     $\hat{o} \leftarrow \text{MajorityVote}(\mathcal{C}(t_p))$    // Get the correct object by majority voting
12     $\mathcal{M} \leftarrow \text{PromptAugment}(t_p, t, s, r, \mathcal{P}, \hat{o}, a, \text{'previous'})$    // Augment prompt by adding above
13     $a_{\text{new}} \leftarrow \text{LLMResponse}(\mathcal{M})$    // Generate or verify answer based on system prompt
14     $a_{\text{ext}} \leftarrow \text{ExtractAnswer}(a_{\text{new}})$
15     **if** $a_{ext} \neq \emptyset$ **and** $a_{ext} \neq a$ **then**
16        $a \leftarrow a_{\text{ext}}$
17     Append $a$ to $\mathcal{A}$
18     Update accumulated prompt $\mathcal{P}$ with $\mathcal{M}$

19  **for** $t_n \in \mathcal{T}_{next}$ **do**            // Process next years after previous years
20     $\hat{o} \leftarrow \text{MajorityVote}(\mathcal{C}(t_p))$    // Get the correct object by majority voting
21     $\mathcal{M} \leftarrow \text{PromptAugment}(t_n, t, s, r, \mathcal{P}, \hat{o}, a, \text{'next'})$    // Augment prompt by adding below
22     $a_{\text{new}} \leftarrow \text{LLMResponse}(\mathcal{M})$    // Generate or verify answer based on system prompt
23     $a_{\text{ext}} \leftarrow \text{ExtractAnswer}(a_{\text{new}})$
24     **if** $a_{ext} \neq \emptyset$ **and** $a_{ext} \neq a$ **then**
25        $a \leftarrow a_{\text{ext}}$
26     Append $a$ to $\mathcal{A}$
27     Update accumulated prompt $\mathcal{P}$ with $\mathcal{M}$

28  **if** $\forall a_i \in \mathcal{A}, a_i = o$ **then**
29     Update knowledge categorization to *Chrono-Correct*

30  **return** $\mathcal{A}$, *Updated Category*

---

## A.8   TASK CONFIGURATIONS OF **CHROKNOWPROMPT**

We apply our method to both *Incorrect* and *Partial Correct* categories, as the latter may still lack definitive answers. The test set consists of 10% of the total dataset from each domain. Evaluation employs fuzzy matching with a temperature of 0 for strict assessment, classifying an answer as *Chrono-correct* only if the last candidate answer matches the object. As described in Section 6.2, the system prompt for each case (generation or verification & refinement) works as follows Table 4:

---

*Generation Case*

**[System]**
Answer 'Candidate A. [Object]' based on the timestamp. Output only the answer: 'A. [Object]'.

---

*Verification & Refinement Case*

**[System]**
Answer 'Candidate A. [Object]' based on the timestamp. If it is correct, repeat the same [Object]. If it is wrong, generate a new [Object]. Output only the answer: 'A. [Object]'.

---

Table 4: System prompts for **ChroKnowPrompt**.

Table 5: Result of **ChroKnowPrompt** for both object changed and unchanged cases. The order of open-sources LLM is sorted by release date, starting from the latest model to the most outdated model. The numeric score represents the level of **Known increase** in chronological categorization, and the increase is due to the transition of previously confusing *Partial correct* responses to *Chrono-correct*. The parenthesis score is the total percentage in dynamic or static dataset, including both changed and unchanged cases. Showing almost 10% to 30% performance of object unchanged cases, the results gives observations that only prompting method has limitations in editing diversely changing temporal knowledge.

| Models | general | | | biomedical | | | legal | | |
|---|---|---|---|---|---|---|---|---|---|
| | dynamic | | static | dynamic | | static | dynamic | | static |
| Object | changed | unchanged | unchanged | changed | unchanged | unchanged | changed | unchanged | unchanged |
| *Proprietary Large Language Models* | | | | | | | | | |
| GPT4o-mini | +0.7 (28.7) | +7.0 (28.7) | +4.7 (33.2) | +1.1 (51.9) | +21.9 (51.9) | +27.8 (51.6) | +0.0 (3.2) | +1.9 (3.2) | +14.1 (51.9) |
| Gemini-1.5-flash | +0.6 (15.6) | +5.9 (15.6) | +4.5 (22.1) | +0.8 (49.0) | +15.0 (49.0) | +16.0 (48.8) | +0.0 (1.3) | +1.0 (1.3) | +14.1 (16.3) |
| *Open-Source Large Language Models* | | | | | | | | | |
| Phi3.5 Mini | +0.3 (17.3) | +1.8 (17.3) | +2.5 (25.5) | +2.1 (45.4) | +16.7 (45.4) | +20.3 (41.3) | +0.0 (0.6) | +0.3 (0.6) | +4.5 (14.2) |
| LLaMA3.1 70B | +0.1 (26.0) | +1.7 (26.0) | +2.1 (33.9) | +1.4 (49.5) | +11.2 (49.5) | +8.7 (46.7) | +0.0 (3.9) | +1.0 (3.9) | +4.5 (56.1) |
| LLaMA3.1 8B | +0.2 (20.6) | +2.8 (20.6) | +1.7 (27.1) | +1.4 (36.9) | +7.8 (36.9) | +7.9 (33.6) | +0.0 (0.3) | +0.0 (0.3) | +1.3 (13.8) |
| Gemma2 | +1.0 (19.6) | +3.0 (19.6) | +2.3 (26.7) | +0.6 (32.5) | +5.6 (32.5) | +9.0 (31.7) | +0.0 (2.9) | +0.6 (2.9) | +2.6 (44.6) |
| Mistral v0.3 | +0.4 (18.6) | +1.5 (18.6) | +1.6 (26.9) | +0.4 (26.6) | +3.8 (26.6) | +5.6 (24.3) | +0.0 (1.3) | +0.6 (1.3) | +7.0 (21.1) |
| LLaMA3 | +0.4 (20.9) | +2.3 (20.9) | +1.7 (28.0) | +0.3 (31.4) | +5.5 (31.4) | +3.8 (25.7) | +0.0 (1.0) | +0.3 (1.0) | +0.6 (18.9) |
| Gemma | +0.2 (18.9) | +0.8 (18.9) | +1.5 (25.9) | +0.3 (18.3) | +5.7 (18.3) | +5.3 (12.6) | +0.0 (0.3) | +0.0 (0.3) | +0.0 (8.70) |
| SOLAR | +0.1 (16.5) | +0.7 (16.5) | +0.9 (24.9) | +0.3 (26.5) | +3.8 (26.5) | +4.5 (20.3) | +0.0 (0.6) | +0.0 (0.6) | +1.3 (26.8) |
| LLaMA2 | +0.3 (18.1) | +4.9 (18.1) | +5.0 (26.6) | +2.0 (44.3) | +23.2 (44.3) | +26.3 (37.2) | +0.0 (0.3) | +0.0 (0.3) | +12.8 (21.8) |
| **Object Increase** | 0.4 | | 2.8 | 1.0 | | 11.6 | 0.0 | | 3.1 |
| **Temporal Increase** | | 1.7 | 2.6 | | 6.0 | 12.3 | | 0.3 | 5.7 |
| **Domain Increase** | | 2.0 | | | 8.1 | | | 2.1 | |

## A.9 DETAILS IN SPAN-WISE RESULTS OF CHROKNOWPROMPT

Table 6 and 7 present the evaluation of **ChroKnowPrompt** in span-wise comparisons. While our approach demonstrates significant improvements in certain domains, it shows limited or negligible gains in the legal domain. Overall scores in the dynamic dataset remain modest, with the highest gain being only 1.9. However, the static dataset yields more impressive results, with the highest increase exceeding 10% in proprietary models, a level comparable to the biomedical domain's results. Another finding is that although the increase in Table 6's result in general domain is not higher than the static figures in the legal domain, the variation in figures between models is significantly larger in the legal domain. As the format of legal dataset is the unstructured format with long context, this would be one factor of low edit quality.

Table 6: Result of **ChroKnowPrompt** in span-wise comparison for general and biomedical domain. The order of open-sources LLM is sorted by release date, starting from the latest model to the most outdated model. The numeric score is the level of **Known** in chronological categorization and the increase in parentheses is from the ratio of *Chrono-correct* which was confusing *Partial correct* before. Each result presents both in total span and previous span.

| Models | general | | | | biomedical | | | | Model Increase | |
|---|---|---|---|---|---|---|---|---|---|---|
| | total span | | previous span | | total span | | previous span | | total span | previous span |
| | dynamic | static | dynamic | static | dynamic | static | dynamic | static | | |
| *Proprietary Large Language Models* | | | | | | | | | | |
| GPT4o-mini | 28.7 (+7.7) | 33.2 (+4.7) | 26.6 (+5.7) | 31.7 (+3.3) | 51.9 (+23.0) | 51.6 (+27.8) | 41.8 (+12.8) | 36.7 (+13.0) | 15.8 | 8.7 |
| Gemini-1.5-flash | 15.6 (+6.5) | 22.1 (+4.5) | 15.3 (+6.1) | 21.7 (+4.1) | 49.0 (+15.8) | 48.8 (+16.0) | 48.0 (+14.9) | 51.7 (+18.8) | 10.7 | 11.0 |
| *Open-Source Large Language Models* | | | | | | | | | | |
| Phi3.5 Mini | 17.3 (+2.1) | 25.5 (+2.5) | 16.5 (+1.2) | 24.1 (+1.1) | 45.4 (+18.7) | 41.3 (+20.3) | 36.6 (+10.0) | 31.5 (+10.5) | 10.9 | 5.7 |
| LLaMA3.1 70B | 26.0 (+1.8) | 33.9 (+2.1) | 26.1 (+1.9) | 33.5 (+1.6) | 49.5 (+12.6) | 46.7 (+8.7) | 44.9 (+7.9) | 41.7 (+3.7) | 6.3 | 3.8 |
| LLaMA3.1 8B | 20.6 (+3.1) | 27.1 (+1.7) | 19.4 (+1.9) | 26.4 (+1.0) | 36.9 (+9.2) | 33.6 (+7.9) | 32.0 (+4.2) | 29.1 (+3.4) | 5.5 | 2.6 |
| Gemma2 | 19.6 (+4.0) | 26.7 (+2.3) | 17.8 (+2.2) | 24.7 (+0.4) | 32.5 (+6.2) | 31.7 (+9.0) | 27.9 (+1.5) | 26.7 (+4.1) | 5.4 | 2.1 |
| Mistral v0.3 | 18.6 (+1.8) | 26.9 (+1.6) | 18.3 (+1.6) | 26.8 (+1.5) | 26.6 (+4.2) | 24.3 (+5.6) | 24.6 (+2.2) | 21.3 (+2.6) | 3.3 | 2.0 |
| LLaMA3 | 20.9 (+2.7) | 28.0 (+1.7) | 20.8 (+2.5) | 27.2 (+0.9) | 31.4 (+5.7) | 25.7 (+3.8) | 28.7 (+3.0) | 24.2 (+2.3) | 3.5 | 2.2 |
| Gemma | 18.9 (+1.0) | 25.9 (+1.5) | 18.8 (+0.8) | 25.3 (+0.8) | 18.3 (+6.0) | 12.6 (+5.3) | 16.0 (+3.7) | 9.60 (+2.3) | 3.5 | 1.9 |
| SOLAR | 16.5 (+0.8) | 24.9 (+0.9) | 16.7 (+1.1) | 25.1 (+1.1) | 26.5 (+4.1) | 20.3 (+4.5) | 27.7 (+5.3) | 19.7 (+3.8) | 2.6 | 2.8 |
| LLaMA2 | 18.1 (+5.2) | 26.6 (+5.0) | 15.9 (+3.0) | 23.1 (+1.5) | 44.3 (+25.2) | 37.2 (+26.3) | 32.5 (+13.4) | 23.3 (+12.4) | 15.4 | 7.6 |
| *Open-Source Chat Models* | | | | | | | | | | |
| Mpt | 18.3 (+4.8) | 25.6 (+4.8) | 17.0 (+3.5) | 22.8 (+2.1) | 43.3 (+22.9) | 45.3 (+30.3) | 30.8 (+10.4) | 26.6 (+11.6) | 15.7 | 6.9 |
| Pythia | 13.8 (+0.0) | 20.8 (+0.1) | 13.8 (+0.0) | 20.7 (+0.0) | 13.1 (+0.0) | 10.2 (+0.1) | 13.1 (+0.0) | 10.2 (+0.1) | 0.1 | 0.0 |
| Nemotron3 | 11.2 (+1.5) | 18.3 (+1.8) | 10.1 (+0.5) | 16.7 (+0.1) | 22.1 (+9.0) | 19.4 (+4.8) | 17.9 (+4.4) | 15.4 (+4.4) | 5.2 | 2.5 |
| **Temporal Increase** | 3.1 | 2.5 | 2.3 | 1.4 | 11.6 | 12.4 | 6.7 | 6.6 | | |
| **Domain Increase** | 2.8 | | 1.8 | | 12.0 | | 6.7 | | | |

Table 7: Result of **ChroKnowPrompt** in span-wise comparison for legal domain. The order of open-source LLMs follows the same sequence as in Table 6, starting with the latest model and progressing to the most outdated one. The numeric score represents the level of **Known** in chronological categorization, and the increase in parentheses reflects the ratio of *Chrono-correct* answers, considering total span in the left side and previous span in the right side.

| Models | legal | | | | Model Increase | |
|---|---|---|---|---|---|---|
| | total span | | previous span | | total span | previous span |
| | dynamic | static | dynamic | static | | |
| *Proprietary Large Language Models* | | | | | | |
| GPT4o-mini | 3.2 (+1.9) | 51.9 (+14.1) | 2.6 (+1.3) | 48.4 (+10.6) | 8.0 | 6.0 |
| Gemini-1.5-flash | 1.3 (+1.0) | 16.3 (+14.1) | 1.6 (+1.3) | 18.5 (+16.3) | 7.6 | 8.8 |
| *Open-Source Large Language Models* | | | | | | |
| Phi3.5 Mini | 0.6 (+0.3) | 14.2 (+4.5) | 0.6 (+0.3) | 11.9 (+2.3) | 2.4 | 1.3 |
| LLaMA3.1 70B | 3.9 (+1.0) | 56.1 (+4.5) | 3.2 (+0.3) | 53.9 (+2.2) | 2.8 | 1.3 |
| LLaMA3.1 8B | 0.3 (+0.0) | 13.8 (+1.3) | 0.3 (+0.0) | 12.5 (+0.0) | 0.7 | 0.0 |
| Gemma2 | 2.9 (+0.6) | 44.6 (+2.6) | 2.6 (+0.3) | 43.9 (+1.9) | 1.6 | 1.1 |
| Mistral v0.3 | 1.3 (+0.6) | 21.1 (+7.0) | 1.0 (+0.3) | 19.2 (+5.1) | 3.8 | 2.7 |
| LLaMA3 | 1.0 (+0.3) | 18.9 (+0.6) | 1.3 (+0.6) | 18.9 (+0.6) | 0.5 | 0.6 |
| Gemma | 0.3 (+0.0) | 8.70 (+0.0) | 0.3 (+0.0) | 8.70 (+0.0) | 0.0 | 0.0 |
| SOLAR | 0.6 (+0.0) | 26.8 (+1.3) | 0.6 (+0.0) | 28.4 (+2.9) | 0.7 | 1.5 |
| LLaMA2 | 0.3 (+0.0) | 21.8 (+12.8) | 0.3 (+0.0) | 17.3 (+8.3) | 6.4 | 4.2 |
| *Open-Source Chat Models* | | | | | | |
| Mpt | 1.0 (+0.6) | 8.4 (+5.1) | 0.6 (+0.3) | 4.5 (+1.3) | 2.9 | 0.8 |
| Pythia | 0.3 (+0.0) | 3.2 (+0.0) | 0.3 (+0.0) | 3.2 (+0.0) | 0.0 | 0.0 |
| Nemotron3 | 0.3 (+0.0) | 5.1 (+1.0) | 0.3 (+0.0) | 4.8 (+0.6) | 0.5 | 0.3 |
| **Temporal Increase** | 0.5 | 4.9 | 0.3 | 3.7 | | |
| **Domain Increase** | 2.7 | | 2.0 | | | |

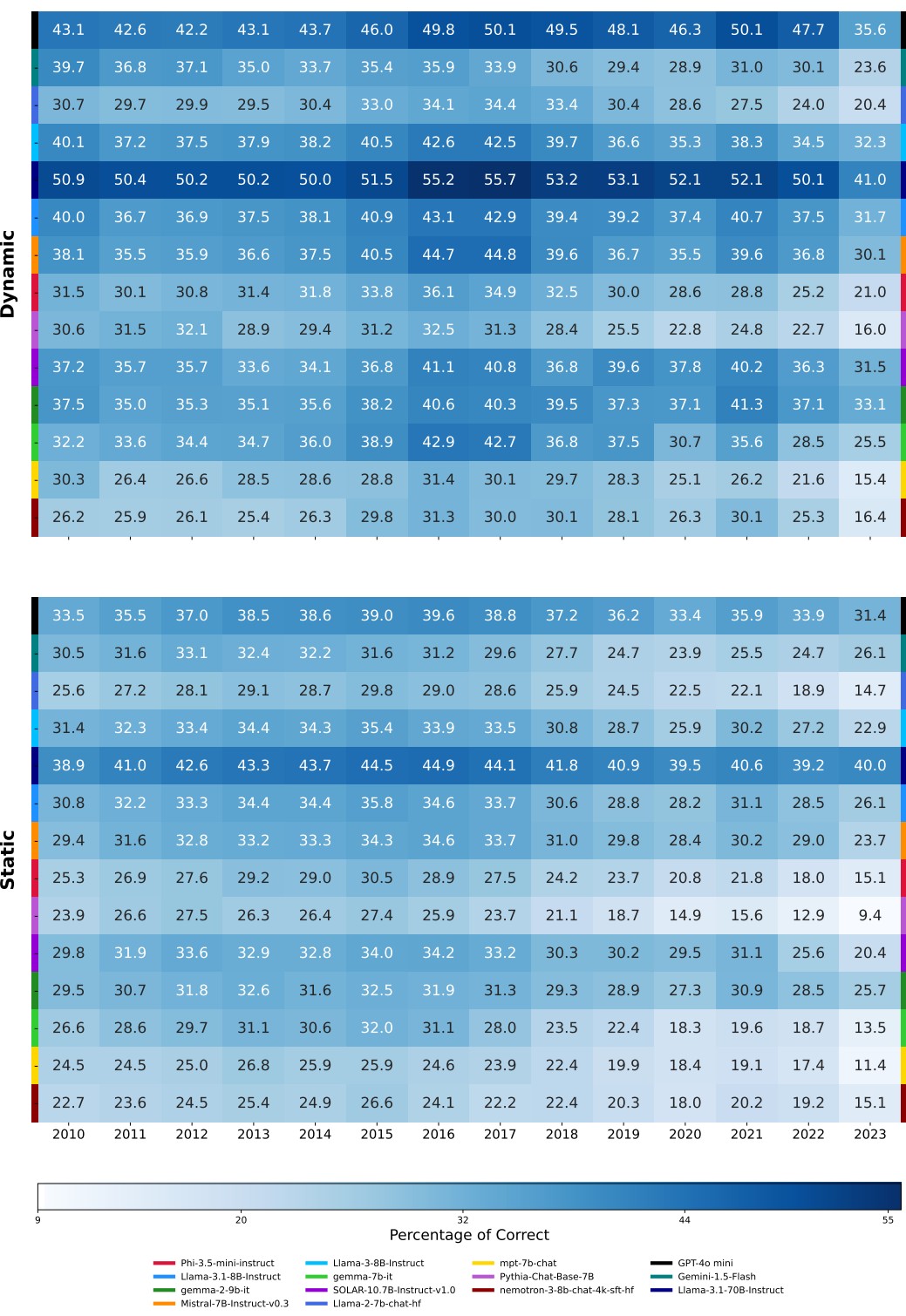

Figure 10: Total result of performance heatmap in general domain for all models

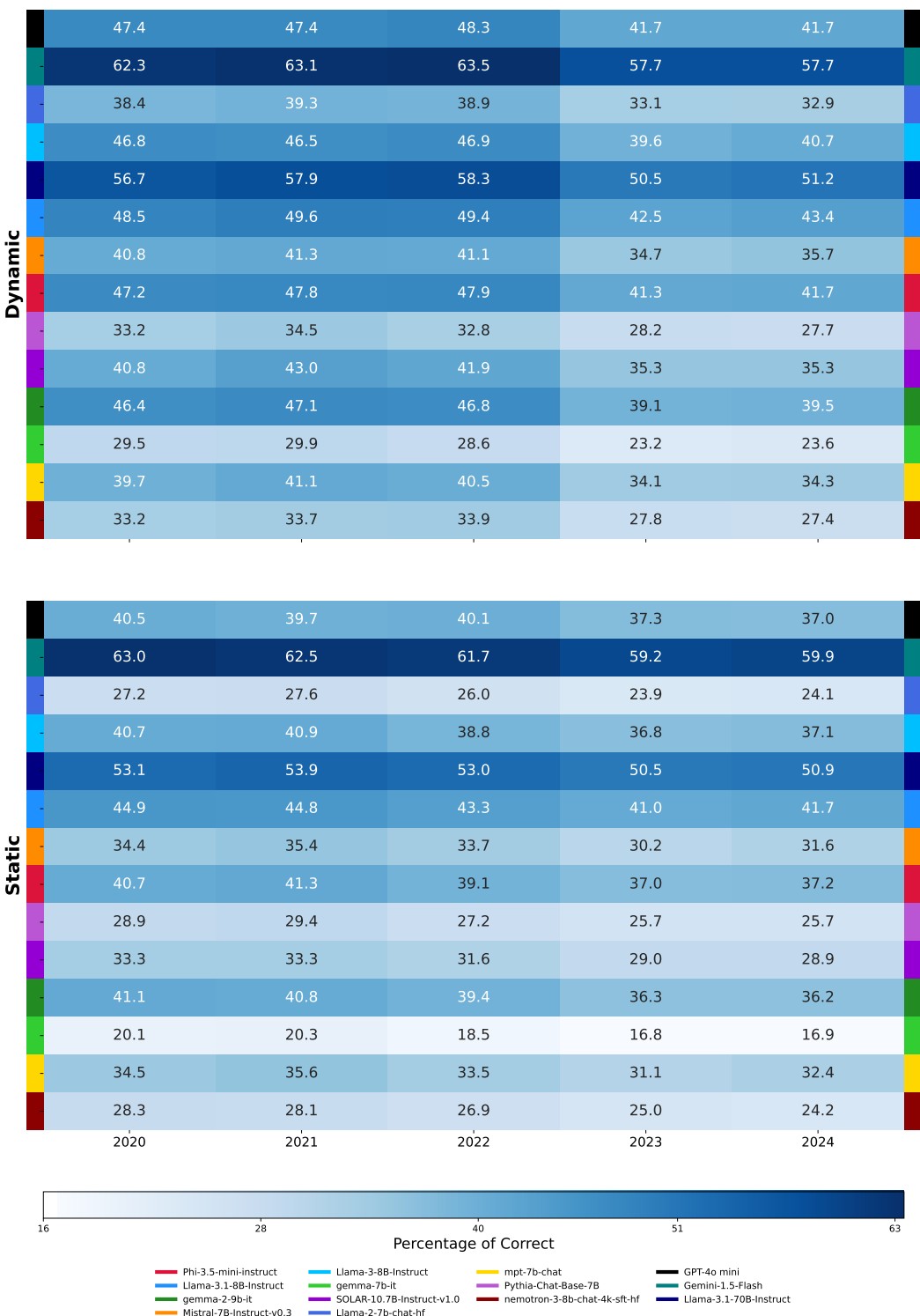

Figure 11: Total result of performance heatmap in biomedical domain for all models

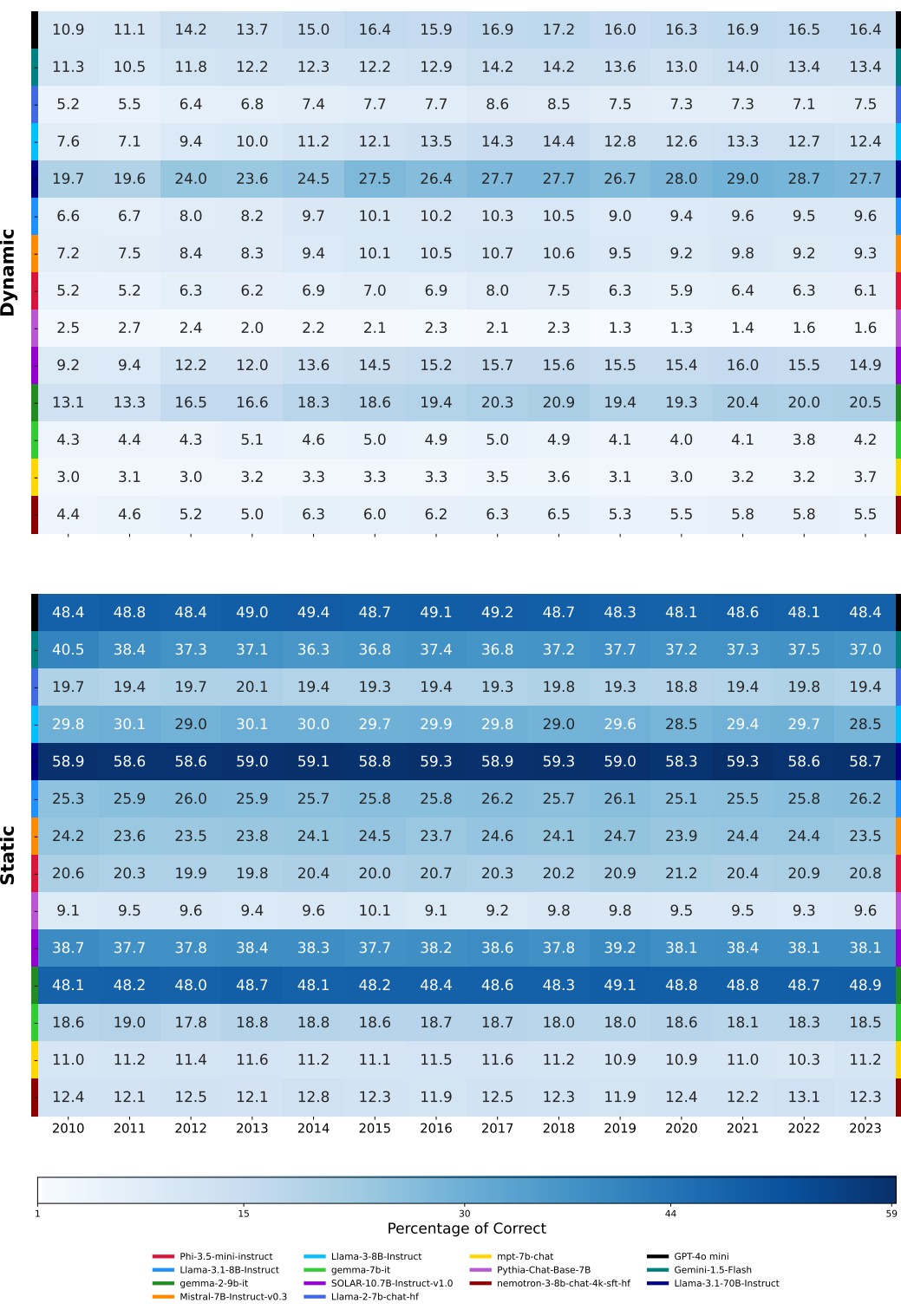

Figure 12: Total result of performance heatmap in legal domain for all models

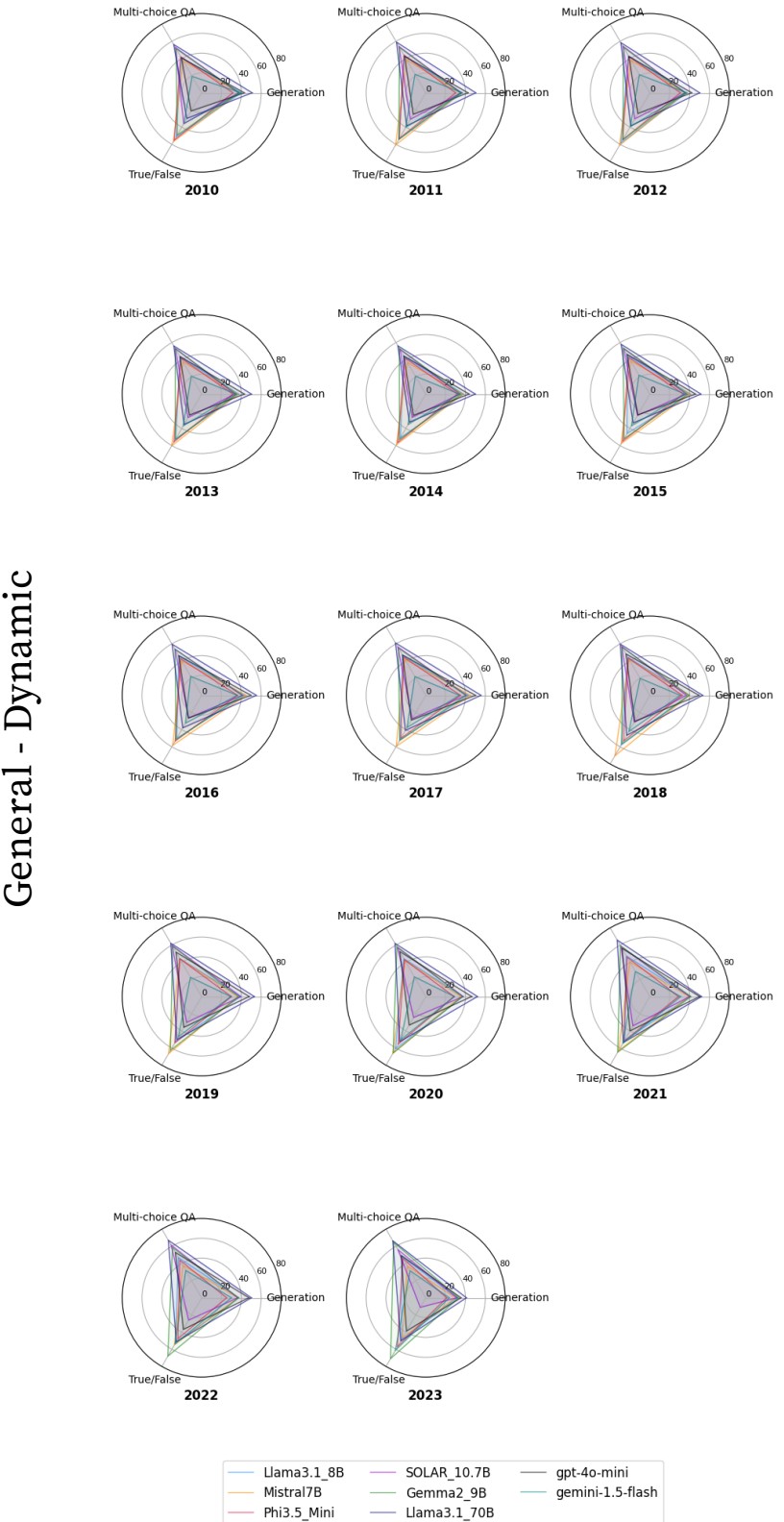

Figure 13: Total result of template-wise performance in general domain, dynamic dataset

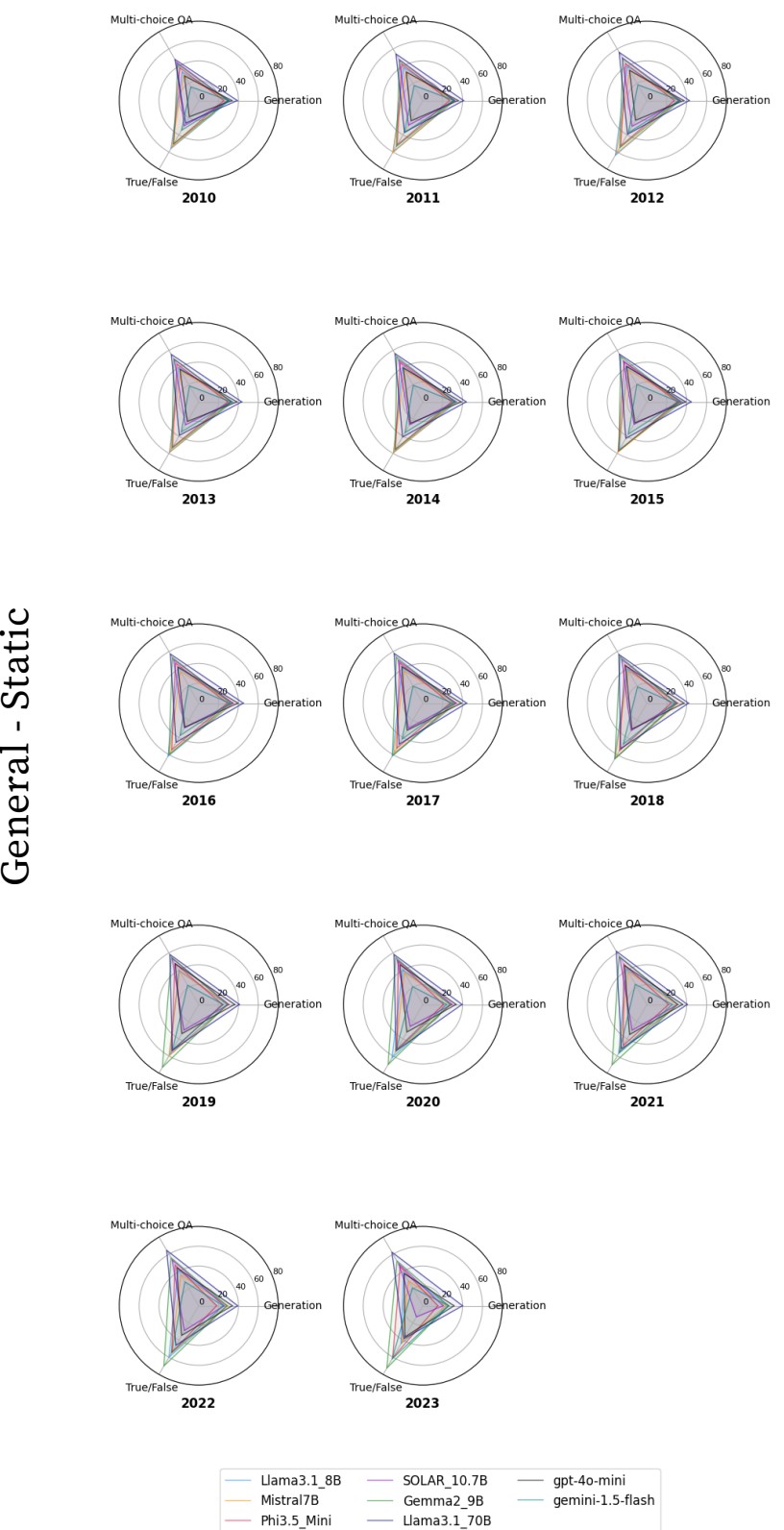

Figure 14: Total result of template-wise performance in general domain, static dataset

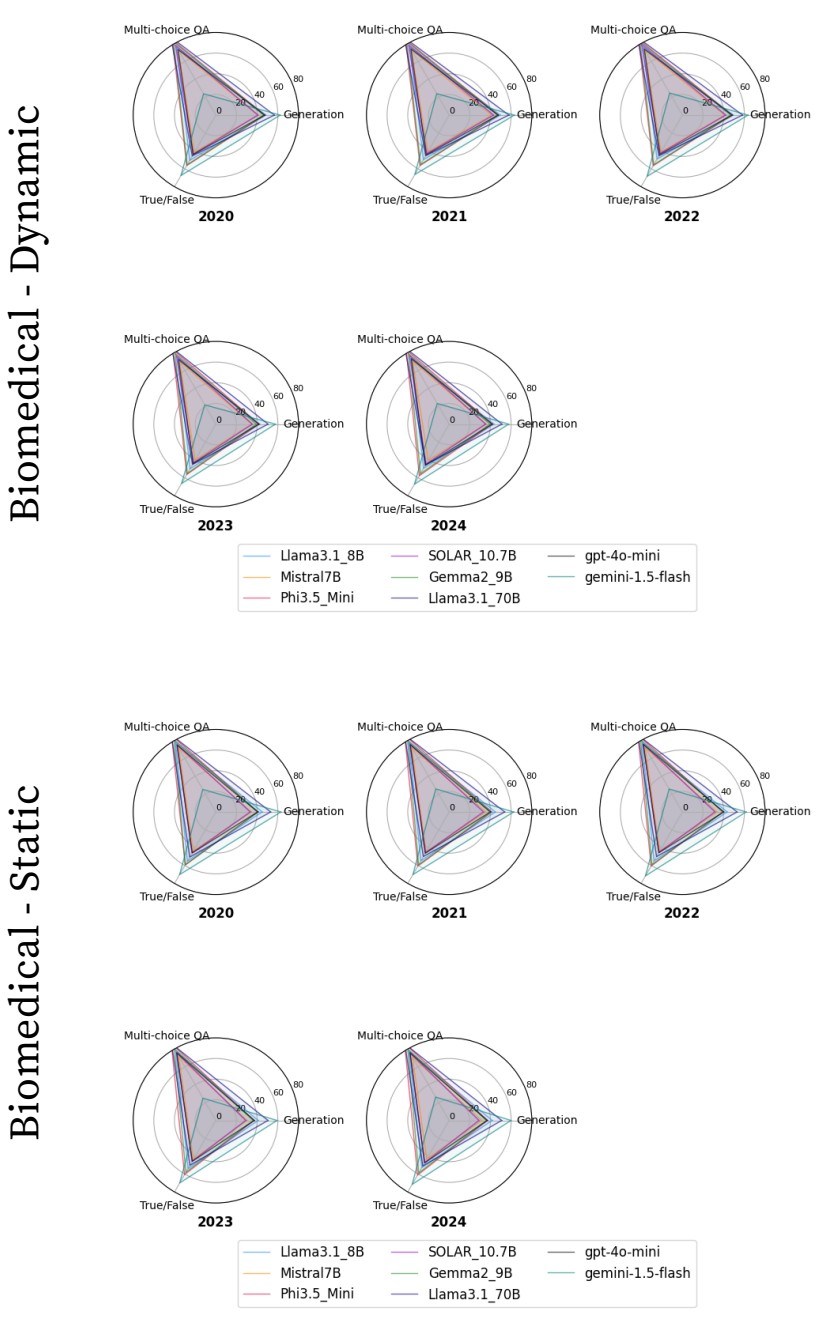

Figure 15: Total result of template-wise performance in biomedical domain, dynamic and static dataset

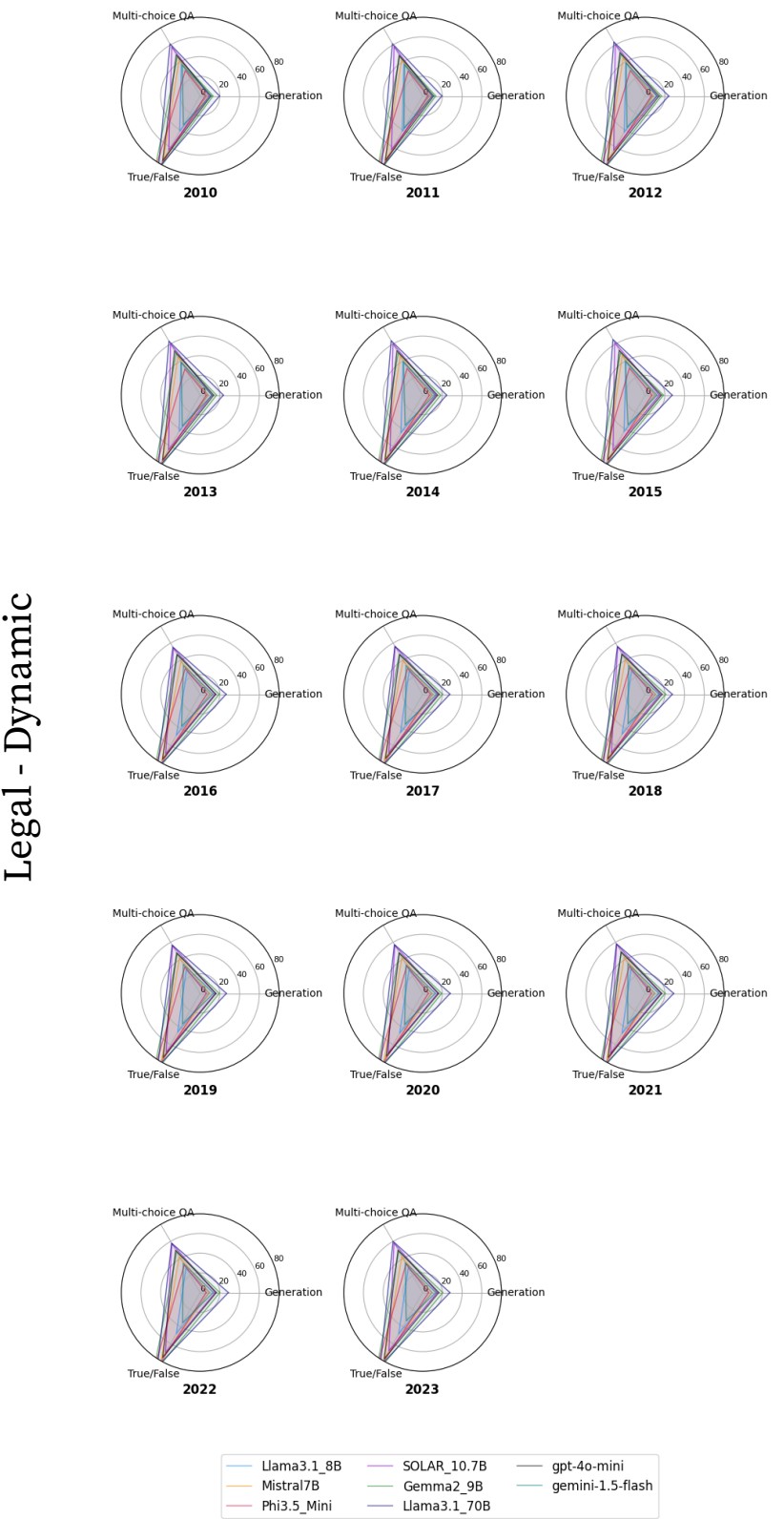

Figure 16: Total result of template-wise performance in legal domain, dynamic dataset

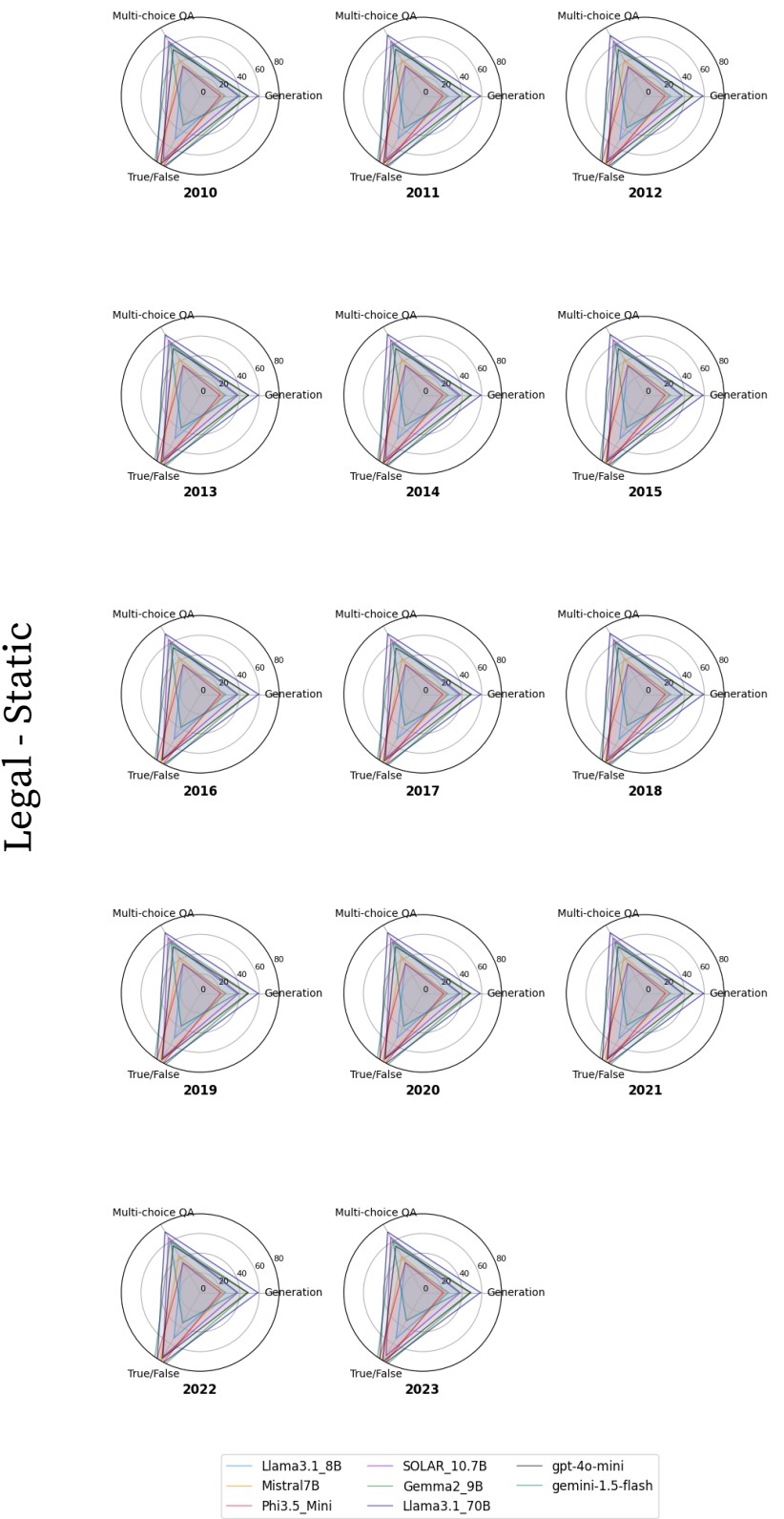

Figure 17: Total result of template-wise performance in legal domain, static dataset

*Template: Generation*

Q. In 2020, Pedro Braillard Poccard, position held, [Object]
A. member of the Argentine Chamber of Senators

Q. In 2020, Ana Santos Aramburo, position held, [Object]
A. Spain National Library general manager

Q. In 2020, James E. McPherson, position held, [Object]
A. United States Secretary of the Navy

Q. In 2020, Jesús Ávila de Grado, position held, [Object]
A. chief scientific officer

Q. In 2020, Donald Tusk, position held, [Object]
: (generate from here ...)

---

*Template: MCQA*

In 2020, what office does Pedro Braillard Poccard hold?
(a) member of the Argentine Chamber of Senators, (b) Minister of Foreign Affairs, (c) Governor of Corrientes Province, (d) Mayor of Buenos Aires

(a) member of the Argentine Chamber of Senators

In 2020, what office does Ana Santos Aramburo hold?
(a) Minister of Culture and Sports of Spain, (b) Director of the Prado Museum, (c) Spain National Library general manager, (d) President of the Spanish Royal Academy

(c) Spain National Library general manager

In 2020, what office does James E. McPherson hold?
(a) United States Secretary of Homeland Security, (b) United States Attorney General, (c) United States Secretary of the Navy, (d) United States Secretary of Defense

(c) United States Secretary of the Navy

In 2020, what office does Jesús Ávila de Grado hold?
(a) President of the National Research Council, (b) Minister of Health, (c) Director of the World Health Organization, (d) chief scientific officer

(d) chief scientific officer

In 2020, what office does Donald Tusk hold?
(a) President of the European Commission, (b) President of Poland, (c) Chancellor of Germany, (d) chairperson

: (generate from here ...)

Table 8: Example of two templates: *Generation* and *MCQA*. Domain: general, Subject: Donald Tusk, Relation: position held (P39), year to target: 2020.

---

*Template: TF*

Q. In 2020, Pedro Braillard Poccard, position held, member of the Argentine Chamber of Senators
A. true

Q. In 2020, Ana Santos Aramburo, position held, Director of the Prado Museum
A. false

Q. In 2020, James E. McPherson, position held, United States Secretary of Defense
A. false

Q. In 2020, Jesús Ávila de Grado, position held, chief scientific officer
A. true

Q. In 2020, Donald Tusk, position held, Prime Minister of Poland
: (generate from here ...)

---

Table 9: Example of *TF* templates. Domain: general, Subject: Donald Tusk, Relation: position held (P39), year to target: 2020.

*Domain: General*

**[System]**
You are an expert in natural language processing and logic puzzles, skilled at generating plausible yet misleading distractor options that challenge users to distinguish between correct and incorrect answers. Pay special attention to questions that involve negative phrasing, such as those containing "not" or "which does not," to ensure that the distractors do not confuse users into overlooking the negative aspect of the question.

**[User]**
The question "What office does 'Mitt Romney' hold?" can be answered with 'United States senator'. Create three plausible incorrect distractors for this question.

**[Assistant]**
1. Governor of Massachusetts
2. Secretary of State
3. Speaker of the House

**[User]**
The question "Which sports team is 'Yann MVila' a member of?" can be answered with 'Rubin Kazan', 'France national association football team', 'Sunderland A.F.C.', 'France national under-21 association football team', 'Inter Milan', 'Stade Rennais F.C.'. Create three plausible incorrect distractors for this question.

**[Assistant]**
1. Paris Saint-Germain F.C.
2. Olympique Lyonnais
3. AS Monaco FC

**[User]**
The question "[Q]" can be answered with '[O1]', '[O2]', ... , '[On]'. Create three plausible incorrect distractors for this question.

**[Assistant]**
: (generate from here ...)

Table 10: 2-shot prompt for generating three distractors in general domain.

*Domain: Biomedical*

**[System]**
You are an expert in natural language processing and logic puzzles, skilled at generating plausible yet misleading distractor options that challenge users to distinguish between correct and incorrect answers. Pay special attention to questions that involve negative phrasing, such as those containing "not" or "which does not," to ensure that the distractors do not confuse users into overlooking the negative aspect of the question.

**[User]**
The question "What is not the primary anatomic site of 'Rhabdomyosarcoma of the orbit'?" can be answered with 'Bones set', 'structure bone', 'Bone structure', 'os', 'Skeletal bone', 'Bone structure (body structure)', 'bones structure', 'bone structures', 'Ossa', 'Set of bone organs', 'Bone organ', 'Skeleton system', 'Osseous', 'Bone', 'NOS', 'Bones', 'bones structures', 'Bone (organ)', 'Bone', 'skeletal', 'Set of bones', 'Bones', 'NOS'. Create three plausible incorrect distractors for this question.

**[Assistant]**
1. Orbital connective tissue
2. Extrinsic ocular muscles
3. Orbital soft tissue

*(two more shots)*

**[User]**
The question "What negatively regulates 'Vitellogenesis'?" can be answered with 'downregulation of yolk production', 'downregulation of vitellogenesis', 'down regulation of vitellogenesis', 'inhibition of yolk production', 'negative regulation of vitellogenesis', 'down regulation of yolk production', 'down-regulation of yolk production', 'inhibition of vitellogenesis', 'down-regulation of vitellogenesis', 'negative regulation of yolk production'. Create three plausible incorrect distractors for this question.

**[Assistant]**
1. Partial left salpingectomy
2. Unilateral oophorectomy
3. Hysterectomy

**[User]**
The question "[Q]" can be answered with '[O1]', '[O2]', ... , '[On]'. Create three plausible incorrect distractors for this question.

**[Assistant]**
: (generate from here ...)

Table 11: 4-shot prompt for generating three distractors in biomedical domain.

---

*Domain: Legal*

**[System]**
You are a legal expert skilled in crafting challenging fill-in-the-blank questions and generating plausible yet misleading distractor options. You will receive a question and answer where part of a legal text has been blanked out. For the provided question, create three plausible incorrect distractors that challenge users to distinguish between correct and incorrect answers.

**[User]**
The question
"""
*... (beginning of text omitted)*
Subpart B—Certification of Substantially Equivalent Agencies
Substantial equivalency certification is granted if the ___ determines that a state or local agency enforces a law that is substantially equivalent to the Fair Housing Act with regard to substantive rights, procedures, remedies, and the availability of judicial review. The Department has developed a two-phase process of substantial equivalency certification.
"""
can be answered with 'Department'. Create three plausible incorrect distractors for this question.

**[Assistant]**
1. Secretary
2. Commission
3. Board

**[User]**
The question
"""
*... (beginning of text omitted)*
Subpart B—Minimum Standards for Substantial Compliance by States
Within the period defined in § 383.73( ___ :
"""
can be answered with 'f) of this title, the State shall', 'h) of this subchapter, the State must'. Create three plausible incorrect distractors for this question.

**[Assistant]**
1. g) of this chapter, the State will
2. e) of this section, the State is required to
3. d) of this part, the State should

**[User]**
The question
"""
`[Q]`
"""
can be answered with '`[O1]`', '`[O2]`', ... , '`[On]`'. Create three plausible incorrect distractors for this question.

**[Assistant]**
: (generate from here ...)

---

Table 12: 2-shot prompt for generating three distractors in legal domain.

*Domain: CommonSense*

**[System]**
You are an expert in natural language processing and logic puzzles, skilled at generating plausible yet misleading distractor options that challenge users to distinguish between correct and incorrect answers. Pay special attention to questions that involve negative phrasing, such as those containing "not" or "which does not," to ensure that the distractors do not confuse users into overlooking the negative aspect of the question.

**[User]**
The question "What properties does 'human' not have?" can be answered with 'strong enough to lift horse', 'strong enough to lift truck', 'strong enough to lift car', 'strong enough to lift cow', 'strong enough to lift building'. Create three plausible incorrect distractors for this question.

**[Assistant]**
1. strong enough to lift a motorcycle
2. strong enough to lift a large suitcase
3. strong enough to lift a heavy stone

**[User]**
The question "What is synonym for 'salix tristis'?" can be answered with 'dwarf gray willow'. Create three plausible incorrect distractors for this question.

**[Assistant]**
1. green weeping willow
2. dwarf green willow
3. gray weeping willow

**[User]**
The question "What does 'kids' desire?" can be answered with 'play games', 'play', 'find crabs in tidal pools', 'read games magazines', 'play with', 'play with toys', 'learn things'. Create three plausible incorrect distractors for this question.

**[Assistant]**
1. eat vegetables
2. do homework
3. fold laundry

**[User]**
The question "What is 'oil and water' not capable of?" can be answered with 'mix'. Create three plausible incorrect distractors for this question.

**[Assistant]**
1. boil
2. freeze
3. emulsify

**[User]**
The question "[Q]" can be answered with '[O1]', '[O2]', ... , '[On]'. Create three plausible incorrect distractors for this question.

**[Assistant]**
: (generate from here ...)

Table 13: 4-shot prompt for generating three distractors in commonsense domain.

---

*Prompt: Mathematics / Data Structure & Algorithm*

**[System]**
You are an expert in mathematics and computer science, skilled at generating plausible yet misleading distractor options that challenge users to distinguish between correct and incorrect answers. Pay special attention to questions that involve negative phrasing, such as those containing "not" or "which does not," to ensure that the distractors do not confuse users into overlooking the negative aspect of the question.

**[User]**
The question "Which of the following is unrelated to 'Output'?" can be answered with 'Left child', 'Set', 'Depth', 'Modify', 'Post-order traversal', 'Dictionary', 'Predecessor', 'Array', 'Delete', 'Sparse matrix', 'Infix expression', 'Leaf node', 'Shortest path', 'Right subtree', 'Node', 'In-order traversal', 'Hashing', 'Head node', 'Pointer', 'Level-order traversal', 'Keyword', 'Pattern string', 'Record', 'Determinism', 'Linked list', 'Critical Path', 'Function name', 'Connected component', 'Loop statement', 'Robustness', 'Inverted index', 'Preorder traversal', 'Memory', 'Dequeue', 'Singly Linked List', 'Image'. Create three plausible incorrect distractors for this question.

**[Assistant]**
1. Print statement
2. Display buffer
3. Output stream

**[User]**
The question "What is synonym for 'Ancestor'?" can be answered with 'Parent'. Create three plausible incorrect distractors for this question.

**[Assistant]**
1. Descendant
2. Sibling
3. Offspring

**[User]**
The question "`[Q]`" can be answered with '`[O1]`', '`[O2]`', ... , '`[On]`'. Create three plausible incorrect distractors for this question.

**[Assistant]**
: (generate from here ...)

---

Table 14: 2-shot prompt for generating three distractors in mathematics, data structure, and algorithm domain.

