# OpenReview forum: "ChroKnowledge: Unveiling Chronological Knowledge of Language Models in Multiple Domains"
_ICLR.cc/2025/Conference — ICLR 2025 Poster_

### Official Review · Reviewer_1hbk · 2024-10-28

**Soundness:** 3
**Presentation:** 4
**Contribution:** 3
**Rating:** 6
**Confidence:** 4

**Summary:**

This paper introduces CHROKNOWBENCH, a benchmark dataset designed to evaluate chronologically accumulated knowledge across three key aspects: multiple domains, time dependency, and temporal state. Besides, the authors develop CHROKNOWPROMPT, an in-depth prompting to elicit chronological knowledge by traversing step-by-step through the surrounding time spans. The motivation behind this work is clear, however, the paper majorly lacks in the quality of the experiments and its setup.

**Strengths:**

This work addresses the sensitivity of LLM to time dependencies and constructs dynamic and static datasets for evaluation.
This paper introduces an iterative approach to integrate knowledge across different temporal spans, enabling dynamic knowledge updates.

**Weaknesses:**

The paper mentions evaluating knowledge across multiple domains; however, it only introduces knowledge from the medical and legal domains, where changes of knowledge along the timeline are relatively subtle.
This paper aims to explore the ability of LLMs to capture temporal dependencies within knowledge. However, the constructed benchmark dataset and evaluation methodology do not effectively demonstrate this capability of the models in an intuitive manner.
Although the paper divides knowledge into dynamic and static datasets, the knowledge within these datasets may have already been learned by LLMs. As a result, the benchmark evaluation results are likely to primarily reflect the memory capacity of the LLM regarding this knowledge, rather than its ability to handle time dependencies.
For the dynamic dataset, this paper essentially transforms knowledge into a temporal knowledge graph (TKG). However, there are established benchmarks in TKGs. The authors should consider providing comparisons with these benchmarks and clarifying the specific improvements made.

**Questions:**

See weakness.

---

> ### Author Response · Authors · 2024-11-21
> **Response to Reviewer 1hbk**
>
> Thank you for your detailed feedback. Your insights help refine our paper and clarify key aspects.
>
> ---
>
> > "The paper mentions evaluating knowledge across multiple domains; however, it only introduces knowledge from the medical and legal domains, where changes of knowledge along the timeline are relatively subtle."
> >
>
> We appreciate this observation and would like to clarify several aspects regarding domain selection and coverage:
>
> 1. **Diverse Domains:**
>     - Our benchmark includes **general knowledge**, which spans multiple temporal knowledge like personal history, and is inherently dynamic. It plays a critical role in evaluating knowledge with frequent temporal changes.
>     - Additionally, we include **time-invariant knowledge**, which provides a baseline for evaluating how temporal cues affect recall across stable datasets.
> 2. **Importance of Medical and Legal Domains:**
>     - Medical and legal domains, while subtle, are pivotal areas where temporal reasoning is crucial for practical applications.
>     - For instance, even minor updates in biomedical guidelines or legal precedents can have significant implications, making their inclusion essential.
>
> **Planned Action:**
>
> - We updated Figure 1 to better emphasize the overall benchmark structures.
> - We plan to expand the dataset to include more diverse and dynamic domains.
>
> ---
>
> > "The knowledge within these datasets may have already been learned by LLMs, primarily reflecting their memory capacity rather than their ability to handle time dependencies."
> >
>
> We agree that demonstrating temporal dependencies distinct from general memory recall is crucial. Our benchmark addresses this through the comparison of **time-variant** and **time-invariant** knowledge with more fine grained analysis inside time-variant; **Dynamic** and **Static**:
>
> 1. **Role of Time-Invariant Knowledge:**
>     - The inclusion of time-invariant datasets helps isolate the effect of temporal cues. While these datasets remain stable regardless of temporal prompts, time-variant knowledge often fluctuates, indicating the influence of time-specific context on recall.
> 2. **Dynamic and Static Comparisons:**
>     - By comparing dynamic and static subsets, we show how temporal cues impact LLM performance. For instance, dynamic datasets reveal where the model struggles to track time-sensitive updates, while static datasets highlight areas of stable recall.
>
> **Planned Action:**
>
> - Revised Figure 2, 3 and 4 shows more clear difference among temporal affected results, with complement view by time invariant results in the Appendix A.5.
>
> ---
>
> > "This paper essentially transforms knowledge into a temporal knowledge graph (TKG). However, there are established benchmarks in TKGs. The authors should provide comparisons with these benchmarks and clarify the specific improvements made."
> >
>
> We recognize the importance of situating our work in the context of existing TKG like HisRES [1]. While there are similarities, our approach differs in significant ways:
>
> - TKG benchmarks frame temporal knowledge in a graph format using temporal snapshot to gather information at that specific time point.
> - In contrast, ChroKnowBench evaluates knowledge through **prompting-based methods**, where temporal context is explicitly provided to test LLMs’ ability to align responses with specific time spans. So our approach is basically based on **Knowledge-Centric Tracking** to construct temporal knowledge.
> - Our temporal prompts enable finer-grained temporal reasoning compared to TKGs, directly query models about specific time points.
> - By testing the integration of multi-temporal knowledge through ChroKnowPrompt, we enable a dynamic assessment of LLMs’ ability to handle temporal dependencies beyond static graphs.
> - We compare TKG datasets in Section 3 and Appendix A.3.3, summarized below.
>
> | **Aspect** | **TKGs** | **ChroKnowBench** |
> | --- | --- | --- |
> | **Temporal Focus** | Temporal snapshots aggregating multiple events | Temporal evolution of individual knowledge elements |
> | **Domain Applicability** | Suitable for well-defined, event-rich domains (e.g., geopolitical, social networks) | Applicable not only to temporal, but also to gradual or implicit changes (e.g., biomedical, legal) |
> | **Handling of Gradual Changes** | Limited by snapshot granularity | Effective through continuous tracking |
> | **Limitations** | May overlook fine-grained changes in individual knowledge | May overlook broader event interdependencies |
>
> **Planned Action:**
>
> - A new subsection compared ChroKnowBench with prominent TKG benchmarks, clarifying the unique contributions of our approach.
> - A table summarizing the methodological differences between ChroKnowBench and existing TKG datasets is included.
>
> ---
>
> We hope these clarifications address your concerns and demonstrate the strengths of our framework. Thank you for your thoughtful and constructive review.
>
> ---
> *[1] Temporal knowledge graph reasoning based on evolutional representation learning*

---

> > ### Comment · Reviewer_1hbk · 2024-11-27
> >
> > Thank you for the detailed response.

---

### Official Review · Reviewer_D8Co · 2024-10-28

**Soundness:** 3
**Presentation:** 3
**Contribution:** 3
**Rating:** 6
**Confidence:** 4

**Summary:**

This paper proposes a new aspect of evaluating Large Language Models, which focuses on chronologically accumulated knowledge. The authors set up a new benchmark, ChroKnowBench, for chronological knowledge evaluation across three dimensions: time dependency, domain difference, and temporal state. They also put forward a novel evaluation framework called ChroKnowledge, and a novel prompting technique called ChroKnowPrompt, which aims to improve the model’s performance using temporal neighboring data.

**Strengths:**

The paper targets a novel field, which studies how language models learn and adapt to knowledge in different time spans. Previously, not many researchers formalized and paid attention to the question. The authors did well in defining the key aspects of testing and evaluating chronological knowledge in large language models, including putting forward a comprehensive dataset and benchmark that covers a variety of time ranges and domains, as well as developing novel prompting techniques. They also conducted fairly comprehensive experiments to demonstrate their claims.

**Weaknesses:**

1.	Results in section 7 does not necessarily support the soundness and validity of the ChroKnowPrompt framework, but instead hints at the deficiency in the ChroKnowledge dataset. There is another possible explanation to the large amount of performance increase in the biomedical section of the dataset: domain-specific data are more static / less dynamic than general data, and your method on distinguishing dynamic and static knowledge needs to be updated to a more fine-grained extent. For example, over the span of 10 years, some data might changed 5 to 6 times, but there could be other data that only changed once. I currently don’t see that taken into account. The improvement from the “dynamic” portion of your results could be over-exaggerated.
2.	The figures need more work. For figure 2 and 3, I understand that the authors are trying to demonstrate the relationship between time and model performance, but the graph needs some extra work on clarity to deliver the message. Figure 4 and 5 in section 6 also needs more work. The most useful explanations for Figure 4 are in the appendix, making the figure a little confusing. Figure 5 needs more clear textual hints on the overall framework of ChroKnowPrompt. The figure itself is not illustrative enough for the entire framework.

**Questions:**

1.	Why would you include Legal data? The rest of the data are of the same type in format and structure, but only legal is unstructured and of a different type.
2.	For your results in section 4, is it possible that the quality of your collected data in different time points is a more influential factor?
3.	Suggestions for the weaknesses mentioned above:
a.	Do more fine-grained classification of the “static” and “dynamic” data.
b.	Update Figure 2 and 3 for more clarity.
c.	Move some information from the appendix to section 7, including the extra descriptions for ChronoPrompt and some extra summaries for the algorithm.

---

> ### Author Response · Authors · 2024-11-21
> **Response to Reviewer D8Co**
>
> Thank you for your detailed and insightful review, which highlights key areas for improvement in our framework and figures.
>
> ---
>
> > "There is another possible explanation to the large amount of performance increase in the biomedical section of the dataset: domain-specific data are more static / less dynamic than general data, and your method on distinguishing dynamic and static knowledge needs to be updated to a more fine-grained extent."
> >
>
> We agree that a more fine-grained distinction between `static` and `dynamic` data would enhance the analysis and provide a clearer understanding of our results.
> - Our current method classifies any entity with one temporal change as dynamic. We analyzed object changes in Appendix A.3.1, quantifying them for results in Figures 2, 3, and 4 (C).
> - Less volatile biomedical data show fewer object changes, aligning with ChroKnowPrompt’s effects on handling unchanged objects.
> - Furthermore, we conducted an ablation study to analyze performance gains based on object changeability in Section 7.2.
> - ChroKnowPrompt performs better with less-frequent changes, as visualized in Figure 7 resulting in the increase of Known.
>
> **Planned Action:**
>
> - We included details in Section 5, 7 and Appendix A.3.1, analyzing the performance of ChroKnowPrompt across varying degrees of dynamicity.
>
> ---
>
> > "Figure 2 and 3 need extra work on clarity to deliver the message. Figure 4 and 5 also need improvement, especially with clearer textual hints and better integration with the main text."
> >
>
> We acknowledge the need to improve the figures for clarity and coherence.
>
> 1. **Figures 2 and 3 → modified in revision Figure 2, 3, and 4:**
>     - Enhanced clarity with heatmap scores, spider plots, and object change quantifications. Results for the legal domain are now included (Figure 4).
> 2. **Figure 4 → modified in revision Figure 5:**
>     - Revised with clearly targeting for ChroKnowPrompt.
> 3. **Figure 5 → Figure 6, Complement with Result in Figure 7:**
>     - More textual explanation in Figure 6, in addition to Figure 7 visualizing ChroKnowPrompt’s impact: notable decrease in Partial Known, increase in Known, and the standout effectiveness in the biomedical domain.
>
> ---
>
> > "The rest of the data are of the same type in format and structure, but only legal is unstructured and of a different type."
> >
>
> We included legal data to address two key goals of our benchmark:
>
> 1. **Diverse Formats:**
>     - The legal domain provides a unique challenge due to its unstructured nature, allowing us to test LLMs' ability to process non-standard formats.
>     - As noted in (Wu et al., 2024b) [1], knowledge editing and recall are particularly challenging in unstructured domains, making this an essential inclusion.
> 2. **Domain-Specific Challenges:**
>     - Legal knowledge often requires nuanced reasoning about temporal and context-dependent facts, making it a valuable addition to our benchmark.
>
> **Planned Action:**
>
> We revised results in Appendix to Figure 4 in revision for including analysis of legal domain, highlighting its unique role in evaluating LLM capabilities for unstructured knowledge.
>
> ---
>
> > "Is it possible that the quality of your collected data in different time points is a more influential factor?"
> >
>
> This is an insightful question. The quality and consistency of data across time points are indeed critical. We took several measures to ensure reliability:
>
> 1. **ChroKnowledge Results:**
>     - For each time point, data were curated to minimize inconsistencies and ensure alignment with the benchmark's goals.
>     - The use of temporal prompts highlights the relevance of knowledge specific to each time point, mitigating biases from unequal data quality.
>     - This is reflected in the results of each (A) in Figure 2, 3, 4.
> 2. **ChroKnowPrompt Results:**
>     - The observed performance improvements in ChroKnowPrompt further validate the importance of multi-temporal data collection, as models perform better when provided with context from multiple time points.
>     - The results visualized with Figure 7.
>
> ---
>
> > "Do more fine-grained classification of static and dynamic data. Update Figure 2 and 3 for more clarity. Move some information from the appendix to Section 7."
> >
>
> We sincerely appreciate these suggestions and agree they will enhance the paper.
>
> 1. **Figure Updates:**
>     - Figures 2 and 3 is redesigned for clarity, and separated with sub plots for fine-grained analysis.
>     - Also, Figure 4 of legal domain originally in the Appendix is added and revised.
>     - Figure for ChroKnowPrompt is complemented with Figure 7 results.
> 2. **Appendix Integration:**
>     - Analysis of ChroKnowPrompt is added in Section 7 with corresponding Appendix.
>
> ---
>
> We believe these clarifications and revisions address your concerns. Your feedback has been vital, and we are confident the revised version will better reflect our contributions.
>
> ---
> *[1] Updating Language Models with Unstructured Facts: Towards Practical Knowledge Editing*

---

> ### Author Response · Authors · 2024-11-26
> **Additional Response to Reviewer D8Co**
>
> Dear Reviewer D8Co,
>
> Thank you for your thorough review of our work. We have been eagerly awaiting your feedback on our previous response and would be grateful to hear your thoughts. Additionally, we would like to share more analysis that addresses your suggestion about the fine-grained classification of data.
>
> ---
>
> > "Do more fine-grained classification of static and dynamic data."
> >
>
> We performed an analysis of the skewness and cumulative characteristics of each domain’s dynamic dataset.
>
> - **Biomedical Domain**
>     - Our analysis shows that the biomedical domain exhibits the least skewness, reflecting a balanced distribution of object changes.
>     - Its characteristic, a narrower range of object change frequencies compared to the general domain, aligns with the observed performance improvements noted in our previous response: ChroKnowPrompt benefits from handling less volatile dynamic datasets, as demonstrated in Figure 7.
> - **General Domain**
>     - The general domain displays a wider range of object changes, though it is more skewed compared to the biomedical domain.
>     - As its range of object change is wider, this variability impacts performance, underscoring the challenge of addressing a domain with greater changeability in object.
> - **Legal Domain**
>     - The legal domain is the most skewed, with the majority of object changes concentrated in a single occurrence.
>     - Despite the narrower range of object change frequencies compared to the biomedical domain, the performance in this domain remains lower.
>     - This suggests that ChroKnowPrompt is more sensitive to the dataset’s format and struggles with unstructured data.
>
> We have included these detailed findings in Figure 8 of Appendix A.3.1, providing comprehensive visualization of skewness and cumulative patterns.
>
> Here, we summarize the statistics of dynamic dataset object changeability.
>
> | Domain | Total Elements | Average Changes | Median Changes | Std Dev | Skewness |
> |--------|----------------|-----------------|----------------|---------|----------|
> | General Domain | 8330 | 2.58 | 2.00 | 1.72 | 1.87 |
> | Biomedical Domain | 7345 | 2.34 | 2.00 | 0.98 | 0.18 |
> | Legal Domain | 3142 | 1.09 | 1.00 | 0.45 | 9.47 |
>
> Although the number of object changes varies across domains, the general domain tends to be more dynamic than others, we believe this effectively represents domain specificity, which contributes to the results of ChroKnowPrompt.
>
> We would greatly appreciate your feedback on whether these additional analyses address your concerns.
>
> If you have any further suggestions for improvement, please feel free to share any additional thoughts or questions.

---

> > ### Comment · Reviewer_D8Co · 2024-11-26
> >
> > Thank you for the detailed response. I have raised the presentation score and would like to keep the remaining scores.

---

> > > ### Author Response · Authors · 2024-11-26
> > >
> > > Thank you for your thoughtful review and for recognizing the improvements in our presentation. We genuinely appreciate how this process has brought attention to the novelty of our work and would greatly value any further insights into areas where additional refinement might be needed. Your feedback has been invaluable, and we remain committed to ensuring the clarity and rigor of our work.

---

### Official Review · Reviewer_P36m · 2024-10-31

**Soundness:** 3
**Presentation:** 3
**Contribution:** 3
**Rating:** 6
**Confidence:** 3

**Summary:**

This paper investigates how well large language models recall chronological knowledge. The paper makes several contributions. First, the authors describe a new benchmark dataset covering different domains (e.g., biomedical, legal) and different types of time dependency and temporal state. Then, the authors benchmark several LLMs (both open-sourced and closed-sourced) and show that LLMs have varied abilities to elicit temporal knowledge. Finally, the authors present a prompting method to elicit chronological knowledge.

**Strengths:**

* The studied problem is important and interesting.
* The authors build a benchmark for evaluating LLM and propose a method for recalling/editing knowledge?

**Weaknesses:**

* The authors evaluate whether LLM can track `object changes`. This problem formulation may not capture the `accumulative nature of knowledge` as discussed in the paper. For example, one drug may have multiple side effects, each of which was identified at different times. I am unsure how the proposed benchmark and evaluation strategy addresses this accumulative nature. For example, a model that always generates the old knowledge will be considered `correct` based on the definition in Table 1.
* The result sections are hard to follow: it is very difficult to understand how their results support these statements in Section 5 (Figures 2, 3).
* The proposed prompting strategy seems to be disconnected from the benchmark sections. For example, the authors show that the proposed prompting strategy improves in the biomedical domain, whereas results in Section 5 and Figure 3 show little variability across times. Any intuitive explanations of why?

**Questions:**

* The ChroKnowBench considers only changes on `object` while keeping `subject` and `relation` unchanged. How does it capture the change of `relation` between the `subject` and `object`? For example, `Zidane` was a `player` at `Real Madrid` in 2001, then became `coach` in 2010.
* Figures 2 and 3: It is hard to interpret these results; suggest using the same y-axis range
* Section 5.1: `In dynamic datasets, models show reduced knowledge in multiple-choice question answering (MCQA) settings, highlighting the impact of formatting and prompting on temporal abilities.` What does this mean?
* Section 5.1: `models fail to track recent updates between 2022 and 2023`; `Static datasets remain consistent, reflecting the stable nature of scientific knowledge` suggest clarifying what these statements mean

---

> ### Author Response · Authors · 2024-11-21
> **Response to Reviewer P36m (Part 1/2)**
>
> Thank you for your detailed review. Your comments highlight areas for clarification and improvement.
>
> ---
>
> > “For example, one drug may have multiple side effects, each of which was identified at different times. I am unsure how the proposed benchmark and evaluation strategy addresses this accumulative nature. For example, a model that always generates the old knowledge will be considered correct based on the definition in Table 1.”
> >
>
> We are wondering if we might not have clearly conveyed the intended scope of our benchmark and evaluation strategy. Our benchmark aims to evaluate whether a model can track the **temporal adaptability** of knowledge at a given time frame, rather than just accumulating all knowledge without temporal awareness.
>
> 1. **How Accumulative Knowledge is Addressed:**
>     - The benchmark evaluates temporal knowledge as it evolves. For instance, if side effects of a drug are identified at different times, the model must provide the correct side effect relevant to the queried time frame.
>     - Here, our methodology is focused on the knowledge first, then we tracked the change of that knowledge across temporal contexts. It is also one of the main difference from Temporal Knowledge Graph(TKG), which is in the Appendix A.3.3.
>     - Generating only outdated knowledge would not be considered correct unless it aligns with the specific temporal context of the query. For example, if model correctly recall old knowledge in appropriate time context, it is correct as it is right in that context.
> 2. **Clarification on Table 1:**
>     - The definitions in Table 1 ensure that knowledge is evaluated based on temporal alignment, explicitly preventing a model from being rewarded for simply repeating outdated information.
>
> **Planned Action:**
>
> We revised our text to more clearly define our concept of `accumulative knowledge` and emphasized how our approach ensures temporal correctness while addressing evolving knowledge. Additionally, we added more detail description compared with other similar datasets.
>
> ---
>
> > “It is very difficult to understand how their results support these statements in Section 5 (Figures 2, 3).”
> >
>
> We acknowledge that the presentation of our results could be improved to ensure clarity.
>
> 1. **Issue with Interpretation:**
>     - The figures may be difficult to follow due to the way the axes are scaled and labeled.
>     - The narrative explaining the connection between the results and the statements in Section 5 could also be more explicit.
>
> **Planned Action:**
>
> 1. We totally modified Figures 2 and 3 for more appropriate explanation with better comparability.
> 2. A clearer explanation of how the results relate to each benchmark dimension (e.g., temporal state, domain) is also provided in Section 5.
> 3. Descriptive captions for figures are expanded to reduce reliance on the main text for interpretation.
>
> ---
>
> > “The proposed prompting strategy improves in the biomedical domain, whereas results in Section 5 and Figure 3 show little variability across times. Any intuitive explanations of why?”
> >
>
> We appreciate the observation and acknowledge the need for a more balanced explanation.
>
> 1. **Clarifying Biomedical Results:**
>     - The biomedical domain exhibits relatively stable performance across dynamic and static datasets, as it is less affected by temporal variability.
>     - While the CHROKNOW Prompt shows some improvement in this domain, its effectiveness appears more pronounced for unchanged objects. This may stem from the biomedical domain's low variability, which allows the model to focus on multi-temporal prompts without excessive interference.
> 2. **Variability Across Times:**
>     - In domains with high temporal variability (e.g., general knowledge), the CHROKNOW Prompt reduces ambiguity for unchanged objects to an extent but faces notable limitations for changed objects. These limitations likely arise from the inherent complexity of dynamic changes, which require more sophisticated temporal reasoning approaches beyond the current prompt design.
>
> By addressing these limitations, our CHROKNOW Prompt suggests as an exploratory step toward handling temporal knowledge though it is not a complete solution.
>
> **Planned Action:**
>
> We explicitly added more explanations in Section 7 with suitable results in Figure 7, emphasizing how temporal variability impacts the effectiveness of our prompting method, which is reducing Partial Known by increasing Known category(completely known in all time frame).

---

> ### Author Response · Authors · 2024-11-21
> **Response to Reviewer P36m (Part 2/2)**
>
> > “The ChroKnowBench considers only changes on object while keeping subject and relation unchanged. How does it capture the change of relation between the subject and object?”
> >
>
> Our benchmark emphasizes **object-level changes** as the primary focus for measuring temporal knowledge dynamics. This decision reflects a balance between practical considerations and the methodological goals of ChroKnowBench. Below, we address this design choice in detail:
>
> 1. **Current Design and its Scope:**
>     - ChroKnowBench evaluates the ability of models to identify and align temporal knowledge for specific **object-level changes** at given time points.
>     - For example:
>         - Query 1: *"In 2001, Zidane was a player at Real Madrid"*
>         - Query 2: *"In 2010, Zidane was a coach"*
>         - These are treated as distinct evaluations to isolate how well the model reasons about **object-level transformations** while keeping the `subject` and `relation` fixed.
>
>     This design captures significant, interpretable changes in knowledge over time while maintaining a structured and scalable evaluation process.
>
> 2. **Why Object-level Focus:**
>     - **Scalability:**
>         - Tracking relational changes creates combinatorial complexity in `subject-relation-object` triples. Object-level focus ensures scalability and clarity.
>     - **Precision:**
>         - Object changes capture nuanced updates (e.g., roles, affiliations) better than relation changes, which are often binary (e.g., *“is a player” → “is not a player”*).
>     - **Flexibility:**
>         - Unlike Temporal Knowledge Graphs which may overlook fine-grained changes in individual knowledge, our object-centered approach avoids rigid relation constraints, enabling precise and interpretable evaluations.
>
> **Planned Action:**
>
> We expanded our discussion in Introduction, Section 3, Appendix A.3.2 and A.3.3 to explain the rationale behind our focus on temporal knowledge along with sufficient comparison.
>
> ---
>
> > “It is hard to interpret these results; suggest using the same y-axis range.”
> >
>
> We understand the concern and agree that consistent y-axis ranges would improve the interpretability of the figures.
>
> **Planned Action:**
>
> 1. We separated same kind of simple line plot into three different kind of plots for its own purpose: heatmap for checking each model’s performance in temporal-wise analysis, spider plot for verifying effect of template in each type of dataset, and bar plot for object changes in dataset in the purpose of detail analysis. Each Figure 2, 3 and additional 4 represents each domain, which becomes more easy as all Figure has same form.
> 2. Annotations highlighting key trends and variations added to aid interpretation in Section 5 and corresponding analysis in the Appendix.
>
> ---
>
> > “What does this mean: models show reduced knowledge in multiple-choice question answering (MCQA) settings, highlighting the impact of formatting and prompting on temporal abilities?”
> >
>
> We apologize for the unclear description regarding the template-wise comparison in the temporal results section.
>
> **Summary of Findings:**
>
> - Before revision, Figure 6 revealed lower and fluctuating scores in dynamic datasets (e.g., legal domain) between 2015–2016 compared to static datasets, suggesting that formatting affects model performance, particularly in specific domains.
> - To address contradictory trends in general and biomedical domains, we introduced an additional template: True/False (T/F).
>
> **Key Insights from Figures 2, 3, and 4:**
>
> 1. Direct answer generation is the most challenging format across domains.
> 2. MCQA and T/F templates complement each other:
>     - **Biomedical domain:** Consistent high performance with MCQA.
>     - **Legal domain:** Significant gains with T/F format.
>
> **Planned Action:**
>
> We rewrote the section and modify figures to clarify the relationship between formatting, prompting, and temporal reasoning, providing insights from our experiments to illustrate the impact.
>
> ---
>
> > “What does this mean: models fail to track recent updates between 2022 and 2023; Static datasets remain consistent, reflecting the stable nature of scientific knowledge?”
> >
>
> We appreciate the opportunity to clarify these statements:
>
> 1. **Dynamic Datasets (2022–2023):**
>     - In Figure 3, models struggle with knowledge recall between 2022–2023 due to the recency of this information and limited exposure during training.
> 2. **Static Datasets:**
>     - These datasets, by design, focus on stable knowledge. Consistent performance in this category indicates that models can recall well-established facts, even without recent updates.
>
> **Planned Action:**
>
> We rephrased these statements in Section 5.1 to better distinguish dynamic and static datasets and their implications for model evaluation.
>
> ---
>
> We hope these responses address your concerns. Your feedback has been essential in improving our work, and we look forward to refining it further. Thank you for your review.

---

> > ### Comment · Reviewer_P36m · 2024-11-25
> >
> > Thank you for the detailed response. I feel that clarifying the scope and main focus makes the paper's contribution much clearer, and the readability has been improved. Therefore, I increased my rating.

---

### Official Review · Reviewer_Yuxp · 2024-11-06

**Soundness:** 3
**Presentation:** 3
**Contribution:** 3
**Rating:** 6
**Confidence:** 4

**Summary:**

This paper introduces CHROKNOWBENCH, a benchmark used to evaluate chronological knowledge in LLMs by distinguishing between evolving and static information. The authors also introduce a sampling-based prompting technique to improve LLMs’ recall of time-dependent knowledge across multiple domains. Their approach enables effective temporal non-parametric knowledge updates, even in proprietary models, showing some improvement in biomedical and general knowledge accuracy.

**Strengths:**

- The paper is very well written, and most of the claims are well-motivated and justified.
- The benchmark covers various domains and temporal knowledge dimensions.
- The authors transform the problem into both open generation and MCQ tasks.

**Weaknesses:**

W1: The overall results appear incomplete across the dataset's various dimensions. A table summarizing average results for each benchmark dimension—Domain, Format, Temporal State, and Time Dependency—would help clarify performance. Currently, it’s unclear whether the performance limitations stem from temporal aspects or the models' domain-specific capabilities.

W2: The proposed prompting method does not clearly demonstrate its effectiveness. Section 5.1 shows stable performance in the biomedical domain, while Section 7.1 suggests the prompting method mainly improves results for biomedical questions, which were not influenced by dynamic changes. How do the authors justify these outcomes as evidence of the prompting method’s efficiency?

W3: In Line 313, the authors state that "GPT-4 performs best in generation tasks." Do they provide any rationale for this observation?

W4: To strengthen the analysis, the paper would benefit from more detailed error analysis and examples. In which cases does the model fail to generate correct answers, and are there discernible patterns in these errors?

**Questions:**

Q1: What is the evaluation score for the "Generation" experiments in Figures 2 and 3? How do you extract the answer from the model output? Is it an exact match?

Q2: You mention that you use the original MMLU prompting strategy (Hendrycks 2021). Is this with the 5-shot setting or with the zero-shot and Chain-of-Thought generation?

Q3: Are the results for the "Time Invariant" data being reported?

---

> ### Author Response · Authors · 2024-11-21
> **Response to Reviewer Yuxp**
>
> We greatly appreciate your insightful feedback, which has been instrumental in improving our paper.
>
> ---
>
> > “A table summarizing average results for each benchmark dimension—Domain, Format, Temporal State, and Time Dependency—would help clarify performance. Currently, it’s unclear whether the performance limitations stem from temporal aspects or the models' domain-specific capabilities.”
> >
>
> We agree that summarizing results across dataset dimensions improves clarity. Current figures visualize trends but lack explicit summaries.
>
> **More clarified observations:**
>
> - Performance of general and biomedical domains decline over time; generation tasks show lowest score but MCQA and TF remain robust.
> - The legal domain performs well on static datasets but struggles with dynamic datasets, indicating a challenge of temporal reasoning in unstructured format.
>
> **Planned Action:**
>
> 1. We reorganized Figures 2 and 3 into three distinct visuals: a heatmap for temporal state performance, a spider plot for format-wise performance over time, and object change statistics.
> 2. Each subsection now includes concise explanations to highlight findings across Temporal State, Format, and Domain.
>
> ---
>
> > “How do the authors justify these outcomes as evidence of the prompting method’s efficiency?”
> >
> 1. **Dynamic vs. Static Knowledge**
>
>     Biomedical data, being less volatile, shows stable results across dynamic and static contexts.
>
> 2. **Object Change Analysis:**
>
>     In Section 7, we analyzed CHROKNOW Prompt’s performance, finding it effective for unchanged objects but less so for changed ones, underscoring its potential and areas for improvement in handling dynamic datasets.
>
> 3. **Why Biomedical Shows Higher Gains**
>     - In the general domain (high variability), gains are limited due to a larger proportion of changed objects.
>     - In biomedical and legal domains, with lower variability, show greater gains in unchanged objects.
> 4. **Demonstrating the Prompt's Efficiency:**
>
>     The CHROKNOW Prompt leverages inherent capabilities for temporal knowledge elicitation without relying on external mechanisms such as retrieval.
>
>
> **Planned Action:**
>
> We clarified these findings in Section 7, supported by Figure 7, and expanded on domain-specific observations.
>
> ---
>
> > “In Line 313, the authors state that ‘GPT-4 performs best in generation tasks.’ Do they provide any rationale for this observation?”
> >
>
> As you noted, GPT-4o-mini (might be mistaken as GPT-4) excels in generation tasks across general domain but is less dominant in other cases. While we don't know details in architecture, its performance in knowledge elicitation suggests substantial exposure to relevant formats during the training stage. Similarly, open-source models like SOLAR excels in MCQA tasks, whereas Gemini-1.5-pro demonstrates weaker performance in this area. Specific formats and data distributions they were exposed to may shape their strengths and weaknesses.
>
> **Planned Action:**
>
> Results are clarified through a new figure and accompanying descriptions.
>
> ---
>
> > “The paper would benefit from more detailed error analysis and examples. In which cases does the model fail to generate correct answers, and are there discernible patterns in these errors?”
> >
>
> We agree that deeper error analysis adds valuable insights.
>
> **Planned Action:**
>
> - Section 7.2 now includes error analysis from three perspectives: Object changeability, Chronological Spans, and Chat Prompting.
> - Supplementary materials in the Appendix provide experimental details.
>
> ---
>
> > “What is the evaluation score for the ‘Generation’ experiments in Figures 2 and 3? How do you extract the answer from the model output? Is it an exact match?”
> >
>
> Evaluation was conducted using a fuzzy matching strategy, as detailed in Section 2.2 and Appendix A.2.2:
>
> 1. Model-generated outputs were parsed and compared with ground-truth object pools using fuzzy match criteria.
> 2. Scores were calculated as the ratio of matches to queries.
>
> **Planned Action:**
>
> While this is mentioned in Section 2.2, we made it more explicit in the Figures 2 and 3 for clarity.
>
> ---
>
> > “Is this with the 5-shot setting or with the zero-shot and Chain-of-Thought generation?”
> >
>
> We apologize for confusion in experimental details. As in Section 2.2 and the appendix:
>
> 1. We used a **4-shot** setting as per MMLU strategy, not 5-shot or zero-shot.
> 2. Chain-of-Thought generation was excluded to focus on direct knowledge checking.
>
> **Planned Action:**
>
> We referenced more papers reflecting this points, in addition to third template: True/False.
>
> ---
>
> > “Are the results for the ‘Time Invariant’ data being reported?”
> >
>
> Yes. The results for "Time Invariant" data are in Appendix A.5 (Figure 9).
>
> **Planned Action:**
>
> Figure 1 and Section 5.1 have been revised to present these findings more prominently.
>
> ---
>
> Thank you for your feedback, which has greatly enhanced our work. We believe these revisions address your comments effectively.

---

> > ### Comment · Reviewer_Yuxp · 2024-11-25
> >
> > Thank you for the detailed response. I have decided to keep my scores.

---

> > > ### Author Response · Authors · 2024-11-25
> > >
> > > We deeply appreciate your thoughtful consideration and feedback. If there are specific areas where you see room for further refinement, we would greatly value your insights. Your guidance is pivotal in helping us enhance the quality and rigor of our work, and we remain committed to continuous improvement.

---

### Author Response · Authors · 2024-11-25
**General Response to All Reviewers**

**Dear AC and Reviewers,**

We deeply appreciate the thoughtful and thorough feedback provided by the reviewers. Overall insights have been invaluable in refining our work, enhancing its clarity, and improving its overall impact. Below, we outline the key revisions and improvements made in response to your comments:

---

### **Revisions and Improvements**

- **Improved Benchmark Presentation and Results:**
    - Figures 2 and 3 have been reorganized into heatmaps, spider plots, and bar charts to better illustrate performance across domains, temporal states, and object changes. Additional captions and contextual explanations were provided to enhance clarity and interpretability. (@Yuxp, @P36m, @D8Co)
    - A fine-grained analysis of dynamic and static datasets has been incorporated in Section 7, quantifying object change frequency and its effect on performance. (@Yuxp, @P36m, @D8Co)
- **Expanded Analysis of Temporal Dynamics:**
    - Results for time-invariant datasets, which provide an essential baseline for evaluating temporal cue effects, have been emphasized more clearly in Section 5, with additional details retained in the appendix for comprehensive reference. (@Yuxp, @1hbk)
    - Domain-specific observations, including stability in biomedical datasets and variability in the legal domain, are now thoroughly discussed in Section 5 to provide greater clarity on temporal knowledge dependencies. (@P36m, @1hbk)
    - We have provided further rationale for observations such as GPT-4's performance in generation tasks, discussing possible reasons related to training exposure to relevant formats. (@Yuxp)
- **Enhanced ChroKnowPrompt Analysis and Error Insights:**
    - Section 7.2 has been expanded to include a detailed error analysis, focusing on object changeability, chronological spans, and template performance. Examples supporting this analysis are provided in the appendix. (@Yuxp, @D8Co)
    - Figures 4 and 5 (now 5 and 6) have been revised to better represent the ChroKnowPrompt framework and its impact. Figure 7 further highlights reductions in partially known categories, demonstrating the effectiveness of the prompting method. (@Yuxp, @D8Co)
- **Clarification on Object-level Changes and Accumulative Knowledge:**
    - We have clarified in the manuscript how our benchmark addresses accumulative knowledge and focuses on object-level changes while keeping subject and relation unchanged, ensuring temporal correctness in knowledge evaluation. (@P36m)
- **Comparison with Temporal Knowledge Graphs (TKGs):**
    - A new subsection and a comparison table in Section 3, Appendix A.3.2, and A.3.3 explain the differences between ChroKnowBench and existing TKG benchmarks, emphasizing finer-grained temporal tracking and broader domain applicability. (@D8Co, @1hbk)
- **Clarifications and Additional Context:**
    - The experimental setup, including details on fuzzy matching and the 4-shot prompting with the new template, has been clarified in Section 2.2 and the appendix. (@Yuxp, @P36m)
    - The inclusion of the legal domain and its unstructured nature has been elaborated, highlighting its importance in evaluating LLMs’ ability to process non-standard formats and temporal reasoning. (@D8Co, @1hbk)
    - We have taken measures to ensure the quality and consistency of our collected data across different time points, addressing potential concerns about data influencing results. (@D8Co)

---

We have carefully revised the manuscript to address these comments, with changes highlighted in blue for ease of reference. The feedback from the reviewers has been invaluable in strengthening the rigor and clarity of our work. We look forward to any further suggestions or comments that may arise.

Thank you again for your thoughtful and constructive reviews.

**Sincerely,**

The Authors

---

### Meta-Review · Area_Chair_F7tB · 2024-12-19

**Metareview:**

This paper introduces a benchmark dataset ChroKnowBench to evaluate chronologically accumulated knowledge of LLMs, and presents a prompting method to elicit chronological knowledge. The problem studies in this paper is important and interesting. The benchmark constructed in this paper may be a valuable resource for future research. Various LLMs have been evaluated on the benchmark. The paper is generally easy to follow, but there are some non-trivial writing issues in the paper that can be addressed by incorporating the content in the authors' responses into the next version.  All reviewers finally hold a positive opinion of the paper, although no one strongly supports it.

**Additional Comments On Reviewer Discussion:**

After the author rebuttal, two reviewers slightly increased their scores and all reviewers hold a positive opinion of the paper, although no one strongly supports it.

---

### Decision · Program_Chairs · 2025-01-22

Accept (Poster)